# Observed different impacts of potential tree restoration on local surface and air temperature

Yitao Li [1,2], Zhao-Liang Li [3] ✉, Hua Wu [4], Xiangyang Liu[3], Xu Lian [5], Menglin Si [3], Jing Li[3], Chenghu Zhou [6], Ronglin Tang[1,2], Sibo Duan[3], Wei Zhao [7], Pei Leng[3], Xiaoning Song[2], Qian Shi[8], Enyu Zhao[9] & Caixia Gao[10]

Tree restoration can cool or warm the local climate through biophysical processes. However, the magnitude of these effects remains unconstrained at large scales, as most previous observational studies rely on land surface temperature (Ts) rather than the more policy-relevant air temperature (Ta). Using satellite observations, we show that Ta responds to tree cover change at only 15–30% of the magnitude observed in Ts. This difference is supported by independent evidence from site observations, and can be attributed to the reduced aerodynamic resistance and the resultant flatter near-surface temperature profiles in forests compared to non-forests. At mid- or high-latitudes, the maximum seasonal biophysical Ta warming or cooling only accounts for approximately 10% of the equivalent climate effect of carbon sequestration in terms of magnitude, whereas the biophysical Ts effect can reach 40%. These findings highlight the importance of selecting the appropriate temperature metric in different applications to avoid exaggerating or underestimating the biophysical impacts of forestation.

In the past decade, the significance of terrestrial ecosystems has gained increasing recognition in high-level climate policies and pledges aimed at combating global climate change[1,2]. A majority of these commitments focus on forest ecosystems[3,4], as global forested areas currently store over 800 petagrams (Pg) of carbon and can absorb ~13 Pg of $CO_2$ from the atmosphere annually[5,6]. Global efforts to reduce the greenhouse effect through forest restoration, known as the biochemical (bchem) feedback of forests, are essential to mitigate global warming[7,8]. Meanwhile, forests present several biophysical (bph)

characteristics, such as lower albedo and greater roughness length, resulting in the local cooling or warming effect compared to their neighboring openlands[9–11]. The sign and magnitude of the local biophysical temperature effects can vary considerably based on spatial location and background climate, and are typically characterized by a shift from cooling effects in the tropics to warming effects in cold regions[12,13]. Forest changes also affect the temperature of spatially nearby regions through advective transport, and even global temperature via altering the large-scale circulation patterns[14]. The

[1]State Key Laboratory of Resources and Environment Information System, Institute of Geographic Sciences and Natural Resources Research, Chinese Academy of Sciences, Beijing, China. [2]University of Chinese Academy of Sciences, Beijing, China. [3]State Key Laboratory of Efficient Utilization of Arable Land in China, Institute of Agricultural Resources and Regional Planning, Chinese Academy of Agricultural Sciences, Beijing, China. [4]School of Resources and Environment, University of Electronic Science and Technology of China, Chengdu, China. [5]Department of Earth and Environmental Engineering, Columbia University, New York, NY, USA. [6]Center for Ocean Remote Sensing of Southern Marine Science and Engineering Guangdong Laboratory (Guangzhou), Guangzhou Institute of Geography, Guangdong Academy of Sciences, Guangzhou, China. [7]Institute of Mountain Hazards and Environment, Chinese Academy of Sciences, Chengdu, China. [8]School of Geography and Planning, Sun Yat-sen University, Guangzhou, China. [9]College of Information Science and Technology, Dalian Maritime University, Dalian, China. [10]Key Laboratory of Quantitative Remote Sensing Information Technology, Aerospace Information Research Institute, Chinese Academy of Sciences, Beijing, China. ✉e-mail: lizhaoliang@caas.cn

magnitude of this nonlocal effect depends on the area extent and the geolocation of the changes[15,16]. Forestation is advocated as an effective solution to achieve the carbon neutrality goal by 2050, and its biophysical feedback can positively or negatively contribute to carbon-related global climate mitigation (the biochemical effect)[17,18].

Currently, forestation practices are predominantly concentrated in limited and specific regions[19,20]. Given that $CO_2$ is well-mixed in the atmosphere, the biochemical feedback on temperature becomes less important when focusing on the climate effects of forestation at regional scales[21]. In contrast, the biophysical effects of forestation can directly induce local cooling or warming, substantially mitigating or exacerbating climate change[22,23]. The mapping of maximum local climate effect through potential forestation practices is informative for policymakers to develop better regional adaptation strategies.

However, existing assessments of large-scale vegetation–climate feedback are subject to various sources of uncertainty. Numerous model-based studies have evaluated the biophysical effects of forest changes under various scenarios over the last two decades[24,25]. Such assessments are dependent on the model representation of surface processes and are biased by the low resolution of simulations[26,27]. High-resolution remote sensing (RS) data provide an avenue for evaluating the potential biophysical effects of forest changes, through a comparison between spatially adjacent forest and non-forest pixels. Nonetheless, for most RS-based studies, the temperature metric is land surface temperature (Ts)[12,28–34], which is a crucial parameter involved in surface energy or water balance processes but has limitations in characterizing the climate effects of forest change. According to the report of the Intergovernmental Panel on Climate Change (IPCC), the indicator used to describe global land warming and frame climate change mitigation targets is land surface air temperature (Ta) rather than Ts[35]. Despite the strong correlation between Ts and Ta[36], the Ts effect of forest change may significantly differ from the Ta effect[37]. Ts-based assessments are useful for model refinement or informing the sign of Ta effect, but the values cannot be directly considered in climate treaties or policies. Although a few studies have explored the different responses of these two temperatures in the context of forest change[38–40], their results may be affected by the uncertainties in numerical models or the sparse distribution of paired forest and non-forest sites. Consequently, it is still unclear whether the biophysical effects of forest change on Ts are comparable with those on Ta at large scales, posing challenges to the direct application of RS-based assessments for policymaking purposes and model result constraints.

This study aims to provide solid observational constraints for the biophysical sensitivity of different temperature metrics to tree cover change and evaluate the impact of potential tree restoration on the local climate. We first estimate the local biophysical Ts and Ta sensitivity to the full tree cover restoration (denoted as $\delta Ts^{bph}$ and $\delta Ta^{bph}$) at the 0.25° scale, based on the space-for-time analogy (Supplementary Fig. 1)[12,41,42]. Notably, the evaluated Ts indicates the radiometric temperature of the land surface, and Ta indicates the air temperature at 2 m above the land surface (Supplementary Fig. 2). The land surface here refers to the interface layer between different land components and the atmosphere (e.g., vegetation canopy)[43]. We revisit previous evaluations of the climate effects of forestation by comparing $\delta Ts^{bph}$ and $\delta Ta^{bph}$, and provide a comparative assessment of the sensitivities across latitudinal, seasonal, and diurnal dimensions. Furthermore, we use the FLUXNET2015 dataset[44] and two gridded temperature datasets to validate the differences between two sensitivities and elucidate the underlying biophysical mechanisms. Finally, we translate the biophysical temperature sensitivities to equivalent $CO_2$ metrics[22], and compare them with the biochemical effects driven by the potential biomass increases, thereby informing the overall climate effects of forest-based climate strategies.

## Results

### Biophysical temperature sensitivities to tree cover gain

The RS-based biophysical sensitivities of annual mean Ts and Ta show similar spatial patterns in terms of sign (Fig. 1a, b). Both $\delta Ts^{bph}$ and $\delta Ta^{bph}$ exhibit positive values in northern high latitudes and negative values in other regions, delineated at around 50°N (Fig. 1c). This spatial distribution reflects a shift from non-radiative cooling in warm regions to radiative warming in cold regions[12,30]. The estimated $\delta Ts^{bph}$ aligns well with a previous study of the potential Ts effect of forestation based on the unmixing method[45], suggesting the robustness to different analytical approaches (Supplementary Fig. 3). In terms of magnitude, $\delta Ta^{bph}$ demonstrates much lower absolute values compared to $\delta Ts^{bph}$ ($-0.14 \pm 0.40$ K vs. $-0.65 \pm 1.22$ K, global mean ± standard deviation), indicating that the local Ta effect of tree restoration is ~22% of the Ts effect. The attenuated $\delta Ta^{bph}$ relative to $\delta Ts^{bph}$ can be observed across all latitudinal bands. At northern high latitudes, the Ta-based warming induced by tree cover change accounts for 32% of the Ts-based warming (0.17 vs. 0.53 K) (Fig. 1d). The ratio of Ta-based cooling to Ts-based cooling is about 40% ($-0.32$ vs. $-0.80$ K) at northern mid-latitudes, 17% ($-0.24$ vs. $-1.41$ K) at tropics, and 23% ($-0.26$ vs. $-1.12$ K) at southern mid-latitude (Fig. 1e–g). These quantitative results are robust to the choice of input tree cover data (Supplementary Fig. 4).

The monthly results further show similar seasonal variation patterns of $\delta Ta^{bph}$ and $\delta Ts^{bph}$, with differing intensities of cooling or warming (Supplementary Fig. 5). In boreal regions, where forest gains predominantly lead to cold season warming effects, the positive monthly $\delta Ta^{bph}$ are considerably lower than $\delta Ts^{bph}$. The ratios of $\delta Ta^{bph}$ to $\delta Ts^{bph}$ in these regions range from 21% to 30%. Conversely, at mid-latitudes, where forestation typically induces a strong growing season cooling effect, ~18% to 33% of the Ts-based cooling can translate into Ta-based cooling. In tropical regions, where forestation results in cooling throughout the year, the negative $\delta Ta^{bph}$ accounts for about 15% of $\delta Ts^{bph}$. These results highlight a consistent pattern in the response of two temperature metrics to forest change, albeit with varying magnitudes.

Previous studies have documented the diurnal asymmetry in the Ts effect of forestation, characterized by cooling at the daytime and warming at the nighttime[12,46]. Our investigation into responses of daily maximum and minimum temperatures to tree cover gain reveals that both the daytime and nighttime Ta effects ($\delta Ta^{bph}_{max}$ and $\delta Ta^{bph}_{min}$) are less pronounced compared to the corresponding Ts effects ($\delta Ts^{bph}_{max}$ and $\delta Ts^{bph}_{min}$) (Supplementary Figs. 6 and 7). Globally, the mean $\delta Ta^{bph}_{max}$ is ~18% of the mean $\delta Ts^{bph}_{max}$ ($-0.41$ K vs. $-2.24$ K), whereas the mean $\delta Ta^{bph}_{min}$ accounts for about 15% of the mean $\delta Ts^{bph}_{min}$ (0.14 K vs. 0.94 K). Across most latitudinal zones, the extent of maximum and minimum Ta sensitivity is notably smaller than that of Ts sensitivity. An exception is observed in tropical nighttime, where the average $\delta Ta^{bph}_{min}$ and $\delta Ts^{bph}_{min}$ exhibit opposite signs ($-0.04$ K vs. 0.13 K) with small absolute values. Overall, we can conclude that roughly 15–30% of the previously observed Ts effects of forest change can translate into climate signals, a proportion that is notably lower than the nearly 50% conversion rate estimated by earth system models[39].

### Validation of the magnitude of Ta sensitivity

Given that the Ta data used for assessment are empirically derived from satellite Ts, rather than direct observations, the accuracy of $\delta Ta^{bph}$ might be dampened by the potential misrepresentations of the Ta retrieval model. To ensure the robustness of our findings, especially the relative magnitude of Ta effects to Ts effects, we further validate the RS-based $\delta Ts^{bph}$ and $\delta Ta^{bph}$ against the temperature effects of forestation ($\delta Ts^{bph*}$ and $\delta Ta^{bph*}$) inferred from the in situ observations and gridded temperature data. Here, $\delta Ts^{bph*}$ and $\delta Ta^{bph*}$ are estimated in different shortwave radiation ($SW_d$) bins to represent the relative

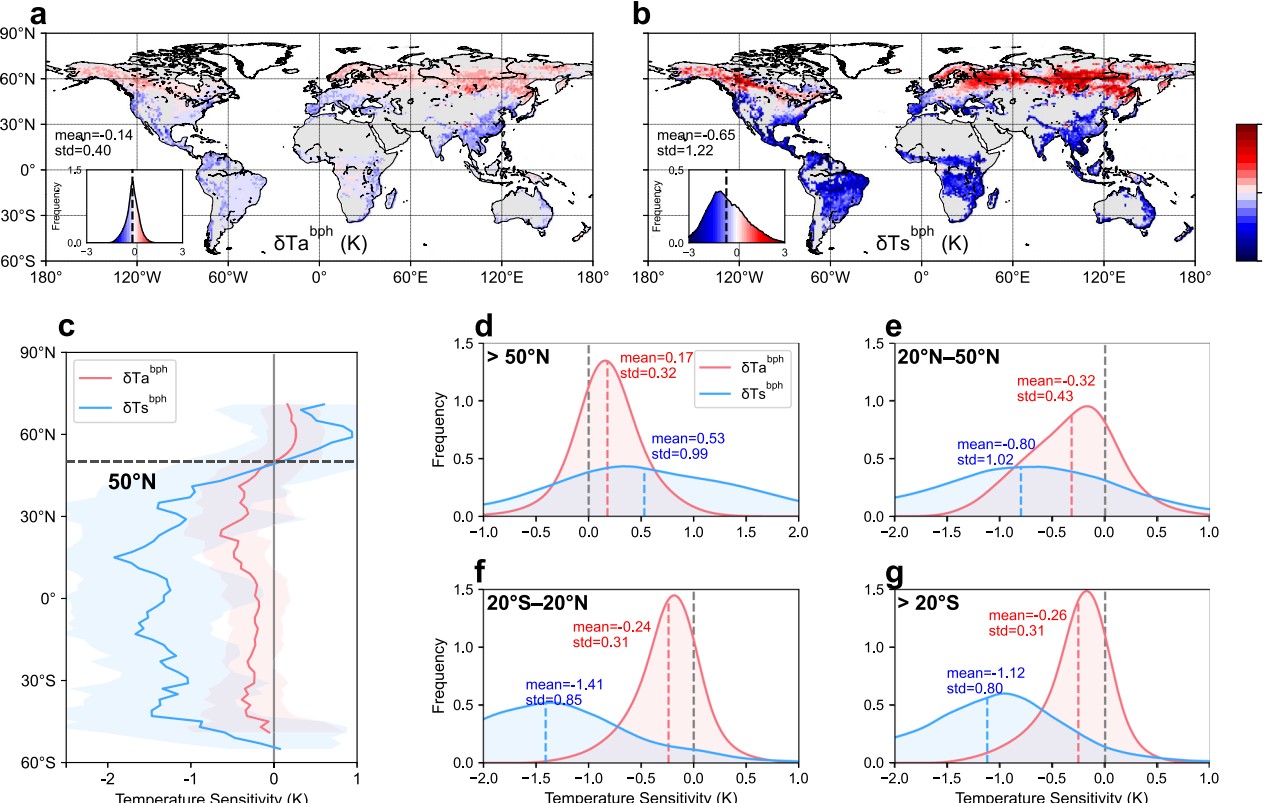

**Fig. 1 | Annual mean temperature sensitivity to the full tree cover restoration.** **a** Global pattern of air temperature sensitivity ($\delta Ta^{bph}$). **b** Global pattern of land surface temperature sensitivity ($\delta Ts^{bph}$). **c** The variation of $\delta Ta^{bph}$ and $\delta Ts^{bph}$ across latitudinal bands, with the shaded area indicating the standard deviation across space. **d**–**g** Probability density of $\delta Ta^{bph}$ and $\delta Ts^{bph}$ across northern high latitudes (>50°N), northern mid-latitudes (20°–50°N), tropics (20°S–20°N), and southern mid-latitudes (>20°S). The tree cover map for the sensitivity estimation is from the GLOBMAP dataset.

changes with changing background radiation conditions (see "Methods" and Supplementary Fig. 8).

The results show that both $\delta Ts^{bph}$ and $\delta Ts^{bph*}$ are negatively correlated with $SW_d$, and the slope obtained from in situ observations matches that derived from RS data (−1.14 vs. −1.39). For the Ta sensitivity, the negative slope derived from in situ observations (−0.24) is also nearly identical to the RS-based results (−0.27) (Fig. 2a, b). The comparable slope values indicate that in situ observations can quantitatively reflect the decrease in temperature sensitivity with increasing radiation, as seen in the RS-based results. Meanwhile, the ratio of slopes indicates that the relative magnitudes of Ta effects to Ts effects are also comparable between the RS-based (19.4%) and in situ results (21.1%).

Since daytime maximum temperatures measure human exposure to heat stress[47], we also validate our findings of maximum Ta and Ts sensitivities via in situ measurements ($\delta Ts^{bph*}_{max}$ and $\delta Ta^{bph*}_{max}$, Fig. 2c, d). The slopes derived from in situ measurements are more pronounced than RS-based results, which may be due to the satellite overpass times (around 13:30, see "Methods") not precisely coinciding with the occurrence of daily maximum temperatures. However, we show that the ratios of Ta sensitivity slopes to Ts sensitivity slopes are close in the RS-based (16.9%) and site-based (17.4%) results (Fig. 2c, d). This result suggests that site measurements corroborate the relative magnitude of the RS-based maximum temperature sensitivity. In addition, we confirm that the validation results are robust irrespective of the choice of gridded temperature data used to control for the impact of macroclimate background (Supplementary Fig. 9). We also perform similar analyses on the site-based minimum Ta and Ts sensitivities ($\delta Ts^{bph*}_{min}$ and $\delta Ta^{bph*}_{min}$, Supplementary Fig. 10), which supports the lower Ta-based warming than Ts-based warming during the nighttime in the RS-

based results. We note that the relationship between $\delta Ta^{bph*}_{min}$ and $SW_d$ is not significant, which corresponds to the weak correlation between $\delta Ta^{bph}_{min}$ and $SW_d$ in the RS-based results ($r = −0.38$). Overall, these results verify the magnitude of the Ta sensitivities derived from the RS data, providing a strong basis for further analysis.

**Biophysical mechanisms of the diverse temperature responses**

To elucidate the biophysical mechanisms underlying the smaller magnitude of $\delta Ta^{bph}$ than $\delta Ts^{bph}$, we also analyze the vertical profile of temperature from the land surface to 2 m height at both forests and non-forested openlands, using the FLUXNET2015 and gridded temperature datasets (see "Methods"). We first focus on winter observations at European sites, which represent high latitudes where forestation leads to dormant season warming. We verify that site observations used in our study can capture the pattern of weaker Ta-based warming than Ts-based warming of forestation (0.16 vs. 1.63 K, Fig. 3a). In such cold environments, the near-surface boundary layer is generally in a stable condition, meaning that the atmosphere tends to warm the land surface, resulting in the temperature inversion phenomenon (Ta >Ts)[48]. The temperature profiles show that the attenuation of $\delta Ta^{bph*}$ in openlands is driven by a more pronounced temperature inversion compared to forests, where Ts is almost identical to Ta (Fig. 3a). Further examination of biophysical property differences reveals that both the absolute values of sensible heat flux (H) and aerodynamic resistance ($r_a$) are greater in openlands than in forests (H: −13.6 vs. −3.4 W·m⁻²; $r_a$: 178.1 vs. 30.7 s·m⁻¹, Fig. 3b). This implies two key factors: first, higher heat flux transfer in openlands favors more pronounced temperature gradients; second, lower transfer efficiency (higher $r_a$) can lead to larger temperature gradients even with constant heat flux. These factors collectively result in more significant

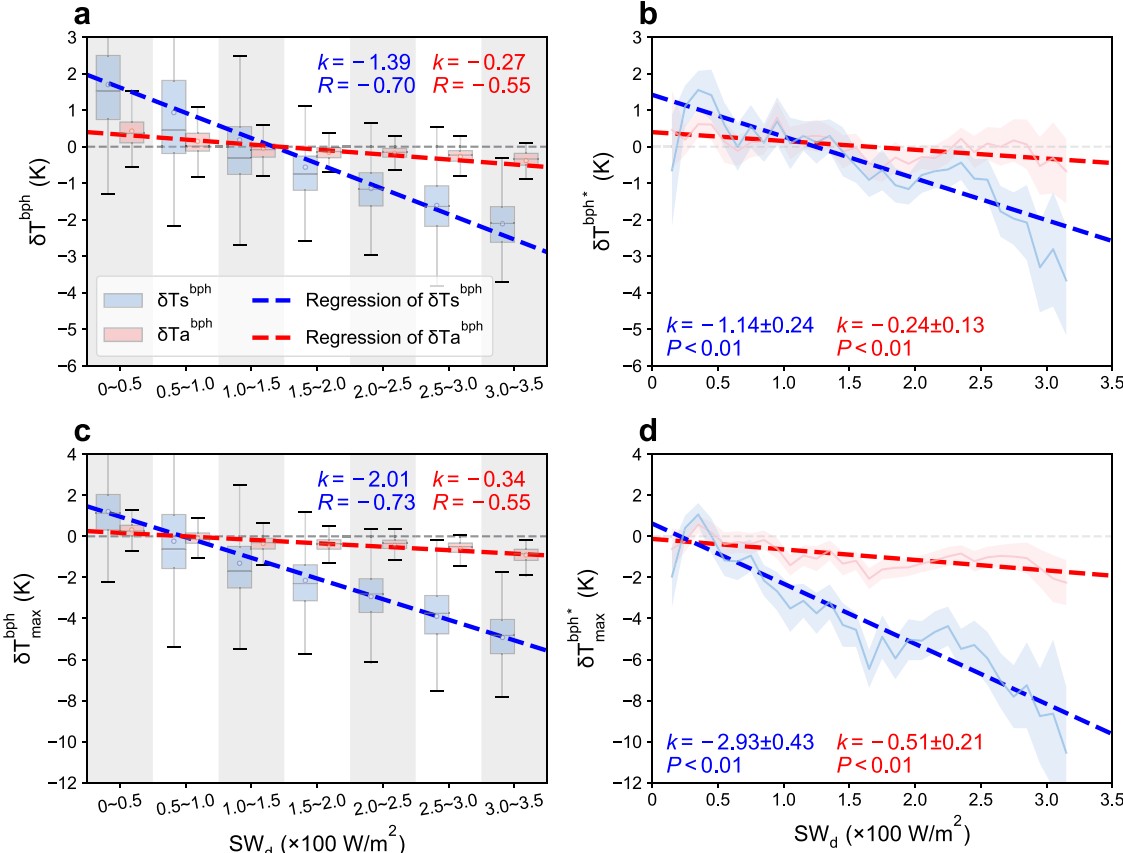

**Fig. 2 | Validation of the monthly land surface temperature and air temperature sensitivities.** **a** Remote sensing-based relationships between mean temperature sensitivities ($\delta Ts^{bph}$ and $\delta Ta^{bph}$) and background shortwave radiation ($SW_d$). The boxplots show the monthly temperature sensitivities within the corresponding $SW_d$ interval. The boxes indicate the interquartile range; the whiskers indicate the data range (5th and 95th percentiles); the lines and dots inside the boxes are the medians and means, respectively. **b** FLUXNET-based relationships between the mean temperature sensitivities ($\delta Ts^{bph*}$ and $\delta Ta^{bph*}$) and $SW_d$. The shaded area indicates the standard error for the mean sensitivity within each $SW_d$ bin. **c** Same as (**a**), but for the maximum temperature sensitivities derived from remote sensing data ($\delta Ts_{max}^{bph}$ and $\delta Ta_{max}^{bph}$). **d** Same as (**b**), but for the maximum temperature sensitivities from FLUXNET measurements ($\delta Ts_{max}^{bph*}$ and $\delta Ta_{max}^{bph*}$). Here, Climatic Research Unit (CRU) temperature data are used to exclude the impact of macroclimate background in FLUXNET temperature measurements.

temperature gradients in openlands (Fig. 3a), thereby contributing to the reduced Ta sensitivity. The quantitative analysis further shows that the impact through $r_a$ ($\delta T^{r_a}$, 55%) slightly outweighs the impact through H ($\delta T^H$, 45%) (Fig. 3c).

We also examine summer observations from North American and Australian sites to understand the mechanisms underlying the reduced Ta cooling in mid- and low-latitude regions. Our results confirm that observations from both regions are consistent with the result of the smaller magnitude of $\delta Ta^{bph*}$ than that of $\delta Ts^{bph*}$ (North America: −0.25 vs. −2.23 K, Australia: −1.13 vs. -3.28 K, Fig. 3d, g). In these warm regions, the land surface is warmer than the ambient air, and the near-surface atmosphere is unstable, characterized by an upward sensible heat flux. The diminished Ta-based cooling effect in forests is attributed to stronger temperature gradients in openlands than in forests (Fig. 3d, g). In terms of the biophysical properties, openlands exhibit higher $r_a$ than forests (North America: 49.2 vs. 9.2 s·m⁻¹; Australia: 45.3 vs. 16.1 s·m⁻¹), whereas the sensible heat flux appears to be similar (Fig. 3e, h). The quantitative analysis also shows that the weaker air cooling is primarily due to forest-resultant decrease of $r_a$ (North America: 82%; Australia: 78%, Fig. 3f, i). Thus, it can be concluded that larger $r_a$ values in openland lead to more pronounced Ts and Ta gradients, resulting in attenuation of the Ta-based cooling effect. These findings confirm the crucial role of $r_a$ in influencing the impacts of both Ts and Ta in response to land cover changes[49]. We note that the contribution of H is greater in European winter than in North American or Australian summer. The possible reason is that H

is more dominant in the turbulent flux exchange during winter (characterized by the higher Bowen ratio) than summer[50], thus contributing more to the temperature gradients between the land surface and the near-surface air, and further to the attenuation of the air temperature response.

## Comparison of biophysical with biochemical effects based on two temperature metrics

Most assessments of the climate benefits related to forestation have concentrated on carbon sequestration (i.e., biochemical effects)[51,52]. Here, the biomass carbon stock sensitivity to tree cover is estimated via space-for-time analogy and converted to $CO_2$ absorption equivalents ($\delta CO_2 e^{bchem}$) to represent the biochemical effect. We also convert the biophysical Ts and Ta sensitivities to the metric of equivalent $CO_2$ uptake ($\delta CO_2 e^{bph, Ts}$ and $\delta CO_2 e^{bph, Ta}$, Supplementary Fig. 11). These allow the comparison of the local biophysical and biochemical climate effects and evaluation of the relative importance of the former[22,23].

The spatial map shows that $\delta CO_2 e^{bchem}$ in tropical rainforest margins can exceed 600 t·ha⁻¹ (Fig. 4a), which is comparable to the previous estimation of tropical intact forest based on ecological research network observations[5]. This value is greater than $\delta CO_2 e^{bchem}$ in temperate and boreal forests, suggesting the highest carbon benefit of restoring damaged or degraded tropical forests. Latitudinally, $\delta CO_2 e^{bchem}$ at low latitudes is higher than that at mid- or high latitudes, with a global mean of 268.2 ± 37.8 t·ha⁻¹ (mean ± uncertainty) (Fig. 4b).

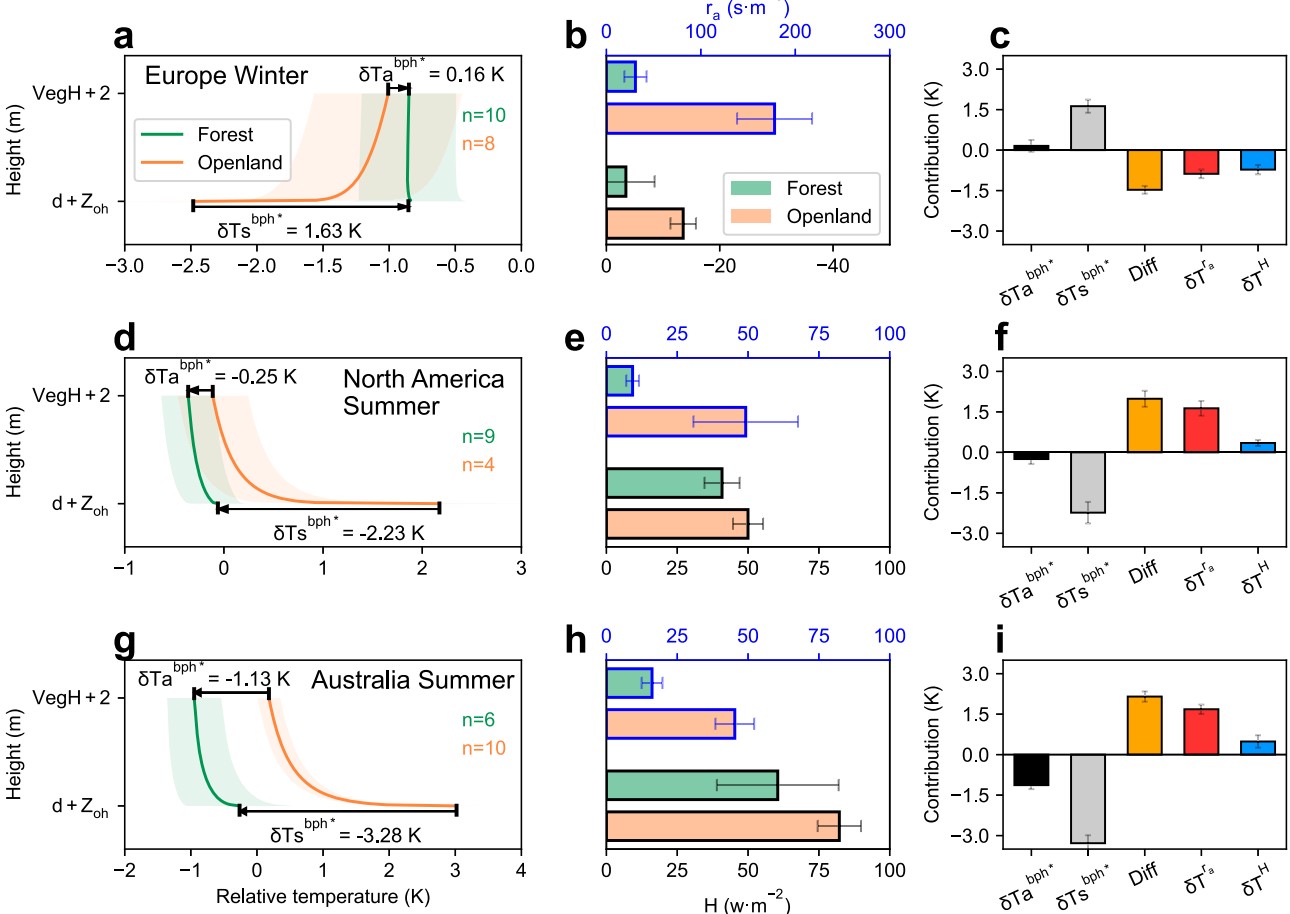

**Fig. 3 | Attribution of differences in temperature sensitivities. a** Vertical evolution of temperature at zero-plane displacement plus heat roughness length (Ts) to the temperature at 2 m above vegetation height (Ta) derived from winter observations of European forest and openland sites. The number *n* at the upper right represents the number of sites, and the shaded area indicates the standard error of multiple-site means. The relative temperature of the *x*-axis is calculated by subtracting the gridded CRU temperature data from the FLUXNET observations. **b** Comparison of estimated aerodynamic resistance ($r_a$) and measured sensible heat flux (H) between forest and openland sites. The error bars indicate the standard error. **c** Bar plots of the mean air temperature sensitivity ($\delta Ta^{bph*}$), land surface temperature sensitivity ($\delta Ts^{bph*}$) and their difference (Diff) contributed by variations in aerodynamic resistance ($\delta T^{r_a}$), and sensible heat ($\delta T^H$). The error bars indicate the standard error. **d–f** Same as (**a–c**), but for summer observations from North American sites. **g–i** Same as (**a–c**), but for summer observations from Australian sites.

In terms of the biophysical effect, $\delta CO_2 e^{bph, Ts}$ (41.7 ± 9.3 t·ha$^{-1}$) provides a global average of 15.7% additional benefits to $\delta CO_2 e^{bchem}$ (Fig. 4b). However, if the more relevant biophysical Ta effect is considered, the ratio of $\delta CO_2 e^{bph, Ta}$ (9.3 ± 2.9 t·ha$^{-1}$) to $\delta CO_2 e^{bchem}$ is only 3.5%.

We then focus on northern high latitudes, where tree restoration shows a biophysical warming effect. The resultant negative $\delta CO_2 e^{bph, Ts}$ could offset 9.5% of the $\delta CO_2 e^{bchem}$ annually (Fig. 4b). The high-latitude biophysical warming is more pronounced in the cold season and can reduce the biochemical climate effect by 42.4% in March (Fig. 4c). However, when $\delta CO_2 e^{bph, Ta}$ is used as the indicator, the offset of biophysical to biochemical effects is only 3.3% at the annual scale, with the maximum monthly value of 10.6% (February) (Fig. 4b, c). In mid-latitudes, the seasonal $\delta CO_2 e^{bph, Ts}$ can enhance $\delta CO_2 e^{bchem}$ by up to 33.7% (northern hemisphere) and 40.5% (southern hemisphere) during summer. However, these seasonal ratios are only about 10% considering $\delta CO_2 e^{bph, Ta}$ (Fig. 4d, f). In low latitudes, annual positive $\delta CO_2 e^{bph, Ts}$ is equivalent to 25.5% of $\delta CO_2 e^{bchem}$, while the ratio for $\delta CO_2 e^{bph, Ta}$ is only 6.2%, with insignificant seasonal variations (Fig. 4b, e). These results suggest that the relative importance of biophysical effects largely depends on the evaluated temperature metric, and the role of biophysical effects in the overall climate effect

(usually measured by Ta) may not be as important as estimated in previous Ts-based studies[22,23].

## Discussion

Previous studies have demonstrated that in boreal regions, forests can warm local Ts because the tree canopy is darker than the snow background and absorbs more solar radiation; in tropical regions, forests show strong local Ts cooling, mainly due to the higher evapotranspiration rates than other vegetation or bare land; in temperate regions, the net Ts effect depends on the relative magnitude of these two processes[10,12,28,30]. However, the Ta effects cannot be simply extrapolated from the Ts effect, as the vertical mixing or coupling between Ts and Ta is much stronger in "rougher" forests than in "smoother" openlands[13,37–39,53]. Leveraging satellite observations, we start by analyzing the biophysical sensitivities of Ta and Ts to tree cover gain. We quantify that ~15–30% of the Ts response could be translated into the Ta response. The less substantial Ta response than the Ts response is validated and further elucidated through in situ measurements, related to the distinct aerodynamic characteristics of forest canopies. Our findings underscore the duality of local Ts and Ta effects induced by tree cover gain, providing a universal metric for translating previous Ts-based results into climate effects. Through the

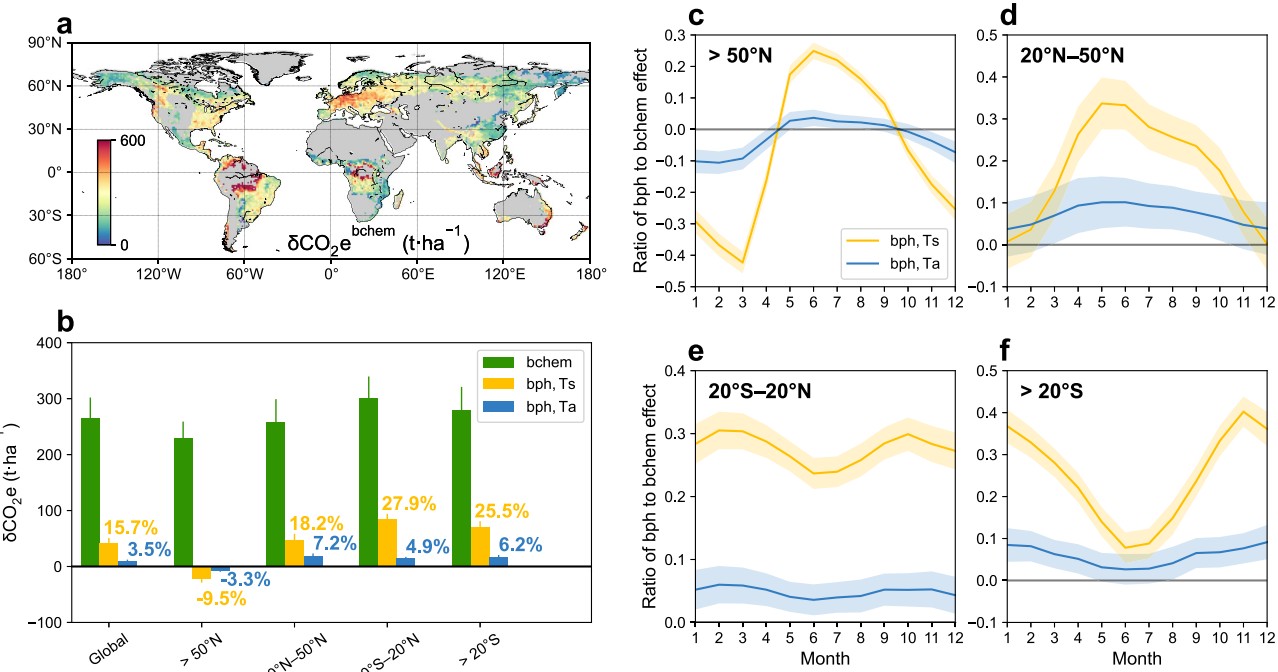

**Fig. 4 | Comparison of the biophysical (bph) and biochemical (bchem) effects of potential tree cover gain. a** Global pattern of the biochemical effect of potential tree cover gain ($\delta CO_2 e^{bchem}$). **b** Global and latitudinal means of biochemical and biophysical effects of potential tree cover gain. The Ts-based and Ta-based biophysical effects are shown as the equivalent $CO_2$ uptake ($\delta CO_2 e^{bph, Ts}$ and $\delta CO_2 e^{bph, Ta}$). The error bars indicate the uncertainty of the mean. **c–f** Monthly ratios of Ta-based and Ts-based biophysical effects to equivalent biochemical effects across northern high latitudes (>50°N), northern mid-latitudes (20°–50°N), tropics (20°S–20°N), and southern mid-latitudes (>20°S). The shaded area indicates the uncertainty of the ratios.

comparison of biophysical and biochemical effects, we find that using Ts as the indicator may overestimate the role of biophysical processes in the overall climate effect of forestation. The evaluation based on the more relevant Ta can present better policy guidance for prioritizing the location of forestation.

The following points should be noted when our results are interpreted. First, akin to prior observational studies, our assessment of biophysical effects does not account for nonlocal or teleconnected effects[25], which could be substantial under scenarios of extensive global tree restoration. For instance, widespread restoration might alter atmospheric circulation patterns[54] and hydrological processes[55] at large or mesoscale scales, thereby affecting the temperature of non-forested areas. The nonlocal effect of forestation can even exceed the local effects in model simulations[15]. Therefore, our estimation of climate benefits should be viewed as the local effect of tree restoration at specific locations. The complex nonlocal feedbacks are better quantified through model simulations, and our findings can serve as constraints for model-based evaluations to more accurately quantify higher-order feedbacks. Second, the estimated biophysical sensitivity of tree restoration is contingent upon current climate conditions, and the impact may evolve in the future. For example, the positive biophysical sensitivity in boreal regions might become negative as snow-induced radiative effects decrease in a warmer world; the impact of rising $CO_2$ levels could also have profound impacts on the climate consequences of forestation[56]. Nonetheless, our observational assessment could be useful for selecting models that better present the biophysical properties of forests and based on which to investigate the climate effects of forestation in future scenarios. Caveats should also be noted for our comparison of biophysical and biochemical effects. Both evaluated biophysical and biochemical effects represent potential cumulative results. It may take a shorter period for biophysical processes (a single decade) to come into effect than biochemical processes, as the newly restored forests gradually absorb $CO_2$ and reach equilibrium after several decades. In addition, potential

variations in soil carbon are ignored in the biochemical part. In contrast to increases in biomass, tree restoration could have positive and negative effects on soil carbon, depending on the climate background and the ecosystem type[57–59]. At the global scale, forestation can increase soil organic carbon, but the value is highly uncertain[60]. Neglecting the potential change in soil carbon may lead to a slight underestimation of the biochemical effect[61]. The evaluated change in biomass should be the main contributor to carbon sequestration[62].

In the context of global climate change, Ts and Ta show comparable variation patterns and trend values[36]. However, when assessing the temperature effects of afforestation or deforestation, the Ts-based values can be about five times higher than the Ta-based values. This significant difference in magnitude highlights that attention should be given to the evaluated temperature metrics and the application scenarios when interpreting the biophysical effects of land cover changes. For instance, Ts (i.e., canopy temperature) could be the more appropriate metric when considering the effects of biophysical processes on ecosystem metabolism of photosynthesis, respiration, and transpiration[37,63]. Meanwhile, the more relevant Ta should be used in analyses related to regional climate adaptation of tree restoration. We highlight that this issue should be considered in future RS-based studies focusing on the thermal buffering effects of forests.

Although Ta-based biophysical effect represents only a small proportion of equivalent biochemical effect, its role in local climate modulation should not be overlooked in regional adaptation strategies. In particular, the diurnal and seasonal changes in biophysical temperature effects should be considered when formulating comprehensive forest-based policies. For instance, we observe maximum temperature cooling and minimum temperature warming effects of forests at high latitudes. This suggests that tree restoration in such cold regions may be a solution to reduce the risks or impacts of daytime warming on the ecosystem. Meanwhile, we show that tree restoration at mid-latitudes can generate considerable summer

maximum Ta cooling, indicating the potential for reducing the impact of hot extremes. Particularly for those mid-latitude countries with ambitious tree restoration goals, tree restoration can offer local climate benefits of mitigating summer heat stress in populated areas.

The biophysical warming effects of boreal forests should be given specific attention in related mitigation policies, although our results of negative biophysical climate effects at high latitudes may not be as strong as previous findings[64,65]. This is because those studies focus on the additional radiative forcing induced by the darker forest canopy but ignore the impact of turbulent fluxes. The overlooked non-radiative effects could partially offset the albedo effects, leading to the observed net warming in our results. From the perspective of the whole climate system, the non-radiative effects represent the redistribution of energy within the climate system and may lead to warming in downwind regions or at the higher boundary layer[66]. Thus, our results concerning the biophysical effects should be treated as the reference for local climate adaptation rather than global climate mitigation. The fact that the mitigation potential of high-latitude forestation could be reduced or even offset by the albedo impacts should be considered by forest-related global policies. Moreover, tree restoration can have numerous ecological, hydrological, and economic impacts besides the assessed temperature effects. Restoration in inappropriate geolocations (e.g., tropical savannas) can have counterproductive consequences such as ecosystem degradation, biodiversity loss, and water availability reduction[67–71]. These impacts should also be considered in the development of comprehensive forest-related strategies to avoid the misconception that "restoring trees is the panacea for the current crisis".

## Methods

### Tree cover map
In this study, two tree canopy cover datasets derived from different sensors are used for the analysis, including the recently released GLOBMAP fractional tree cover map with a spatial resolution of 250 m[72], and the Global Forest Change (GFC) tree cover map with a spatial resolution of 30 m[73]. Considering the data availability, we use the tree cover maps of 2010 ($TC_{2010}$) of both two products to calculate the biophysical temperature sensitivities to ensure robustness. Both $TC_{2010}$ maps from GLOBMAP and GFC are preprocessed and spatially aggregated to the 1 km resolution for further analysis (Supplementary Fig. 12).

### Satellite-based Ts and Ta
The thermal infrared sensors onboard satellites provide direct measurements of Ts. In forested land, Ts represents the mixture temperature of the tree canopy and the exposed soil at the observed angle. Here, the monthly mean Ts data of 2010 are generated by the daily four observations from Moderate Resolution Imaging Spectroradiometer (MODIS) onboard Terra and Aqua satellites (observed at 1:30, 10:30, 13:30, and 22:30). Specifically, the four instantaneous Ts observations (MOD11A1/MYD11A1) are first converted to the daily mean values using the weighted average method[74], and the daily values are then temporally aggregated to monthly mean values[75]. The synthesized monthly Ts data are the all-sky average with the spatial resolution of 1 km, and show satisfactory accuracy compared to the in situ measurements (root mean square error of 1.6 K). The daily maximum and minimum Ts data are calculated from the mean values of MODIS observations (13:30 and 1:30) at the monthly scale.

The other temperature metric used for assessment is Ta, the air temperature at ~2 m above the interface layer between the land components and the atmosphere. Specifically, for forested areas, the reference plane is the canopy, whereas for openlands, the reference plane is approximately the ground (Supplementary Fig. 2). Here, we use a state-of-the-art spatiotemporal seamless Ta dataset to analyze the local climate effect of potential forestation[76,77]. This dataset is derived from a statistical model that correlates Ta from about 100,000

weather station records with satellite Ts (observed at 13:30 and 1:30) and other auxiliary variables. It provides global 1 km daily maximum and minimum air temperature data, with accuracies of -2 K and -1.5 K, respectively. We first aggregate the daily data to the monthly scale. Then, the monthly mean Ta data are calculated by arithmetically averaging the monthly maximum and minimum Ta values. Satellite monthly mean Ta and FLUXNET monthly mean Ta show good agreement. The validation results for forest and non-forest sites show comparable accuracy (Supplementary Fig. 13).

### Calculation of biophysical temperature sensitivity maps
The biophysical sensitivity in this study is defined as the potential local temperature change when tree cover increases from 0 to 100%. A positive (or negative) sensitivity value at a given location indicates a local warming (or cooling) effect due to full restoration. This sensitivity is estimated using the space-for-time method[12,53], which assumes that the spatial variability of Ts or Ta within a designated area reflects land surface property differences, given that pixels within this area share the same macroclimate. Specifically, for each $0.25° × 0.25°$ grid cell[23], we filter out pixels with more than 1% water body coverage or less than 10% tree cover according to the forest definition by the Food and Agricultural Organization[78]. This process is to reduce the impact of non-forest land cover types on the estimation of temperature sensitivity. We also exclude pixels with elevation differences exceeding 100 m from the average elevation of the 0.25° grid to avoid the potential impact of altitude on temperature. The water coverage and elevation data are from Joint Research Center Global Surface Water Mapping Layers v1.4[79] and GMTED2010 datasets, respectively.

After the screening process, $δTs^{bph}$ and $δTa^{bph}$ can be estimated using a linear regression model between tree cover and corresponding temperature values for each 0.25° grid[13,80,81] by Eqs. (1) and (2):

$$Ts = δTs^{bph} × TC_{2010} + b_s \tag{1}$$

$$Ta = δTa^{bph} × TC_{2010} + b_a \tag{2}$$

where, $b_s$ and $b_a$ are the regression intercepts. To ensure the reliability of the results, biophysical sensitivity calculation is performed only when the total sample size of the linear regression model exceeds 90 (more than 10% of pixels within the 0.25° grid) and the difference between the highest and lowest tree cover is greater than 40%. $δTs^{bph}$ and $δTa^{bph}$ are calculated using monthly data from 2010, and the extreme 1% values at both ends are removed from each sensitivity map to exclude outliers. The annual sensitivity is then averaged from these monthly results. In addition to mean temperature, we also calculate sensitivities for maximum and minimum temperatures using the same method, thereby exploring the diurnal temperature effects of forestation in more detail. Notably, all the sensitivity results should be interpreted as the temperature consequences of restoration with native forest type, as the gridded tree cover data of existing species are used as inputs to the spatial regression model.

### Validation of RS-based biophysical sensitivities
Previous model-based and site-based studies have shown that Ts and Ta exhibit distinctive responses to forestation or deforestation processes at various scales[38–40]. Here, our RS-based quantitative analysis also shows that the Ta effect is considerably weaker than the Ts effect. However, it is important to note that the Ta data employed for calculating biophysical sensitivity are derived from a statistical model using satellite Ts observations as inputs, rather than from direct space-based measurements. This raises the possibility that the calculated $δTa^{bph}$ might be influenced more by uncertainties inherent in the Ta statistical model across different land cover types, rather than

accurately reflecting the true Ta effect of tree cover change. Therefore, there is a need to corroborate the magnitude of $\delta Ta^{bph}$ and the relative ratio of the two sensitivities using additional evidence.

The RS-based local temperature sensitivity can be validated through the differences in measurements between spatially adjacent paired forest and non-forest sites. However, the sparse spatial distribution of such paired sites is insufficient to support global-scale validation of biophysical sensitivity[34,82]. Inspired by the methodology proposed in a previous study[83], we use comprehensive flux tower measurements with the requisite variables from the monthly FLUX-NET2015 Tier 1 dataset (Supplementary Table 1), along with interpolated air temperature data from the Climatic Research Unit (CRU TS4.06) and Berkeley Earth Surface Temperatures (BEST)[84,85] to validate the results. The method is based on assumptions that interpolated air temperature data primarily reflect macroclimate conditions and is, therefore, less sensitive to land cover; while the in situ measurements reflect the both impacts of land cover and macroclimate climate. Compared with traditional paired analysis, this methodology enables us to analyze the effects of Ts and Ta due to land cover changes, leveraging spatially distant tower data or temporally asynchronous observations.

The specific process of validation is as follows (Supplementary Fig. 8). In situ data for Ta are measured above the vegetation canopy, whereas Ts is estimated using the longwave radiation measurements[86] by Eq. (3):

$$Ts = \left[\frac{LW_u - (1-\varepsilon)LW_d}{\varepsilon\sigma}\right]^{\frac{1}{4}} \tag{3}$$

where, $LW_u$ and $LW_d$ represent upward and downward longwave radiation from the FLUXNET2015 dataset, respectively; $\sigma$ denotes the Stephan–Boltzmann constant ($5.67 \times 10^{-8}$ W m$^{-2}$ K$^{-4}$), and $\varepsilon$ is emissivity, estimated based on an empirical relationship with albedo[87]. For the gridded data, we first make corrections using the lapse rates to compensate for the elevation difference between the site and the corresponding grid. The lapse rate for the target grid is estimated by the regression slope of the gridded temperatures and elevations within the $5 \times 5$ window.

By deducting the corrected gridded temperature data, the in situ measurements can effectively represent the land cover impacts on local Ts and Ta, assuming that macroclimate affects both temperature metrics similarly. Since the forest data cannot be directly matched with the openland data, we bin both forest and openland data points using the $SW_d$ interval of 10 w·m$^{-2}$. For each $SW_d$ bin, we calculate the difference between mean values of forest and openland data points to represent the temperature effect of forestation (i.e., $\delta Ts^{bph*}$ or $\delta Ta^{bph*}$) under the specific radiation background using Eqs. (4) and (5):

$$\delta Ts^{bph*} = \overline{\left(Ts_f^{site} - T_f^{grid}\right)} - \overline{\left(Ts_o^{site} - T_o^{grid}\right)} \quad if \; SW_d \in (10k, 10k+10) \tag{4}$$

$$\delta Ta^{bph*} = \overline{\left(Ta_f^{site} - T_f^{grid}\right)} - \overline{\left(Ta_o^{site} - T_o^{grid}\right)} \quad if \; SW_d \in (10k, 10k+10) \tag{5}$$

Here, $Ts_f^{site}$ and $Ta_f^{site}$ refer to Ts and Ta measured at forest sites, respectively; $Ts_o^{site}$ and $Ta_o^{site}$ refer to Ts and Ta measured at openland sites; $T_f^{grid}$ and $T_o^{grid}$ refer to the corresponding gridded temperatures after the elevation correction; $k$ indicates counting of the $SW_d$ bin. According to the metadata of the FLUXNET2015 dataset, forest sites include the following four IGBP land cover types: evergreen needleleaf forests, evergreen broadleaf forests, deciduous broadleaf forests, and mixed forests; openland sites are categorized as other non-forest vegetation types.

Then, the relationships between two temperature sensitivities and $SW_d$ are explored using the weighted least squares (WLS) regression model, in which the samples are $\delta Ts^{bph*}$ or $\delta Ta^{bph*}$ of all $SW_d$ bins and the sample weights are defined as the inverse of the standard error of $\delta Ts^{bph*}$ or $\delta Ta^{bph*}$. The derived relationships are then compared with those from RS-based results for validation. Here, the monthly ERA5-Land shortwave radiation data are used to build the relationships with RS-based sensitivities. We also compare and validate the maximum and minimum temperature sensitivities.

### Analysis of near-surface temperature profiles

To further investigate the biophysical mechanism behind the varying magnitudes of $\delta Ts^{bph}$ and $\delta Ta^{bph}$, we estimate and compare the vertical evolution from Ts to Ta in forest and openland sites. This comparison analysis uses daily meteorological, turbulence, and radiation records from sites in North America, Europe, and Australia (Supplementary Table 1). Specifically, we focus on winter observations from European sites (or summer observations from North American or Australian sites) to examine the typical magnitude differences in air and land surface warming (or cooling) effects of forestation.

Here, we first normalize the Ta measurements to the theoretical values at the 2-meter above the vegetation canopy to exclude the potential impact of measurement heights on the results[38]. This normalization process is based on the parametrization of aerodynamic resistance ($r_a$) using the Monin–Obukhov similarity theory[88]. Specifically, the theoretical relationship between Ts (the extrapolated temperature value at the height of heat roughness length plus zero-plane displacement) and Ta (the measured temperature above the canopy at height z) can be expressed by Eqs. (6) and (7):

$$Ta(z) = Ts - \frac{Hr_a(z)}{\rho C_p} \tag{6}$$

$$r_a(z) = \frac{1}{0.4u^*}\left[\ln\left(\frac{z-d}{Z_{oh}}\right) - \psi_h\left(\frac{z-d}{L}\right)\right] \tag{7}$$

where $\rho$ is the air density, $C_p$ is the specific heat of air at constant pressure, d is the zero-plane displacement and is assumed to be 67% of the vegetation height[89], $u^*$ is the friction velocity, $Z_{oh}$ is the heat roughness length, and $\Psi_h$ indicates the stable correction for heat, which is the function of the Monin–Obukhov length (L)[88]. $Z_{oh}$, the only unknown parameter required to solve the temperature profile, is determined under certain conditions[38]: (1) H has the same sign as Ts−Ta; (2) the absolute value of H exceeds 20 w·m$^{-2}$; (3) $u^*$ is greater than 0.01 m·s$^{-1}$; and (4) the atmospheric stability parameter ($\frac{z-d}{L}$) falls between 1 and -2[38]. Invalid roughness length values are then filled using the relationship between the logarithm of inferred $Z_{oh}$ and the friction velocity[38,90].

With the $Z_{oh}$ inferred for both forest and openland sites, Ta can be estimated by modifying z to 2 m above the vegetation canopy in Eq. (6). Then, the impact of potential afforestation on Ta and Ts can be derived by comparing the near-surface temperature profiles of forest and openland sites. Similar to the sensitivity validation approach, we exclude the impact of background climate using the corresponding gridded air temperature data to ensure the comparability of the measurements. Meanwhile, through the first-order expansion of the analytical expression for Ta, we can decompose the air temperature sensitivity into the contributions from two biophysical parameters ($\delta T^H$ and $\delta T^{r_a}$) using Eq. (8), given the known sensitivity at $Z_{oh} + d$ ($\delta Ts^{bph*}$):

$$\delta Ta^{bph*} = \delta Ts^{bph*} + \delta T^H + \delta T^{r_a} \tag{8}$$

Here, $\delta T^H$ and $\delta T^{r_a}$ are calculated by the partial derivatives ($\frac{\partial T_a}{\partial H}$ and $\frac{\partial T_a}{\partial r_a}$) and the parameter difference between forest and openland ($\delta H$

and $\delta r_a$, defined as forest minus openland) using Eqs. (9) and (10):

$$\delta T^H = \frac{\partial Ta}{\partial H}\delta H \qquad (9)$$

$$\delta T^{r_a} = \frac{\partial Ta}{\partial r_a}\delta r_a \qquad (10)$$

This decomposition process allows a quantitative evaluation of why the absolute Ta response to forestation is smaller than the Ts response. Specifically, if $\delta T^H$ is dominant, it implies that the differing sensible fluxes for heating or cooling the near-surface atmosphere are responsible for the milder air temperature response; if $\delta T^{r_a}$ is more significant, it suggests that changes in heat convection efficiencies, leading to different steepness in the temperature profiles, contribute to the attenuation of the air temperature response.

## Comparison of biophysical and biochemical effects

In addition to regulating the energy balance process, forestation can enhance the land carbon sink through vegetation photosynthesis, thereby generating negative biochemical feedback on the climate system[91]. To quantify this biochemical impact, we first estimate the biomass carbon density sensitivity to ideal restoration, using Global Aboveground and Belowground Biomass Carbon Density Maps of 2010 (in $t \cdot ha^{-1}$)[92], along with $TC_{2010}$ and the "space-for-time" strategy. We convert the biomass carbon stock sensitivity to $CO_2$ absorption equivalents (i.e., $\delta CO_2 e^{bchem}$) based on the molar mass ratio. Notably, $\delta CO_2 e^{bchem}$ provides a simple estimate of the ideal carbon stock in biomass under current climate and disturbance regimes for further comparison with the biophysical effect. The period for restored forests to reach such carbon potential, as well as the role of changing climate and soil carbon flux in this process are neglected.

The biophysical Ts and Ta sensitivities are also unified to the metric of $CO_2$ equivalents ($\delta CO_2 e^{bph, Ts}$ and $\delta CO_2 e^{bph, Ta}$) using Eqs. (11) and (12), based on the transient climate response to cumulative emissions for both Ts ($TCRE^{Ts}$) and Ta ($TCRE^{Ta}$) derived from Coupled Model Intercomparison Project Phase 6 (CMIP6) simulations (Supplementary Fig. 11):

$$\delta CO_2 e^{bph, Ts} = \frac{\delta Ts^{bph}}{TCRE^{Ts}} \times \frac{1}{A_E} \qquad (11)$$

$$\delta CO_2 e^{bph, Ta} = \frac{\delta Ta^{bph}}{TCRE^{Ta}} \times \frac{1}{A_E} \qquad (12)$$

where, $A_E$ indicates the earth surface area ($5.1 \times 10^8$ km$^2$). The gridded $TCRE^{Ts}$ and $TCRE^{Ta}$ are estimated following the previous study[22], using 12 model simulations (ACCESS_ESM1-5, CanESM5-1, CMCC-ESM2, CNRM-ESM2-1, FIO-ESM-2-0, GISS-E2-1-H, INM-CM5-0, IPSL-CM6A-LR, MIROC6, MPI-ESM1-2-LR, MRI-ESM2-0 and NESM3) of the "1 percent per year increase in carbon dioxide" experiment (1pctCO2). In the calculation, we consider that 1 ppm of atmospheric $CO_2$ corresponds to 7.82 gigatonnes $CO_2$ and assume that the airborne fraction of the $CO_2$ flux is 43%[13], as the 1pctCO2 experiment is based on the increase in $CO_2$ concentration, rather than the emission. Notably, $\delta CO_2 e^{bph, Ts}$ and $\delta CO_2 e^{bph, Ta}$ calculated by Eqs. (11) and (12) represent the $CO_2$ emission equivalents. We further convert their signs to align with the $\delta CO_2 e^{bchem}$, which represent the $CO_2$ absorption equivalents. We compare the biophysical and biochemical effects based on the above metrics at both annual and monthly scales.

## Data availability

All the data that support the findings of this study are openly available. GLOBMAP fractional tree cover can be downloaded from https://zenodo.org/records/10589730. GFC tree cover data are available at https://glad.umd.edu/dataset/global-2010-tree-cover-30-m. MODIS land surface temperature data are available at https://ladsweb.modaps.eosdis.nasa.gov/search/. Satellite-based air temperature data can be downloaded from https://iastate.figshare.com/collections/A_global_1_km_resolution_daily_near-surface_air_temperature_dataset_2003_2020_/6005185. The Joint Research Center Global Surface Water Mapping Layers are available at https://developers.google.com/earth-engine/datasets/catalog/JRC_GSW1_4_GlobalSurfaceWater. GMTED2010 Elevation data are available at https://developers.google.com/earth-engine/datasets/catalog/USGS_GMTED2010. CRU gridded temperature data can be downloaded at https://catalogue.ceda.ac.uk/uuid/e0b4e1e56c1c4460b796073a31366980. BEST gridded temperature data can be downloaded at https://berkeleyearth.org/data/. ERA5-Land reanalysis data are available at https://cds.climate.copernicus.eu/cdsapp#!/dataset/reanalysis-era5-land-monthly-means. Global Aboveground and Belowground Biomass Carbon Density Maps are available at https://developers.google.com/earth-engine/datasets/catalog/NASA_ORNL_biomass_carbon_density_v1. FLUXNET2015 dataset is available at https://fluxnet.org/. CMIP6 simulations can be downloaded from https://esgf-node.llnl.gov/search/cmip6/.

## Code availability

The Python codes used to generate all the results are available at https://zenodo.org/records/14633331.

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

## Acknowledgements

This work was supported by the National Natural Science Foundation of China (Grant No. 41921001, Z.-L.L.; 42071331, H.W.), the Strategic Priority Research Program of Chinese Academy of Sciences (Grant No. XDA28050200, H.W.). The authors acknowledge constructive and insightful comments from Prof. Ronggao Liu and Prof. Yang Liu. The authors give our sincere thanks to all data providers for their continuous efforts and sharing the data.

## Author contributions

Y.L., Z.-L.L., and H.W. conceived and designed the research. Y.L. and X. Liu organized and processed the data. Y.L., X. Liu, M.S., and J.L. carried out the analysis. Y.L. drafted the paper. Z.-L.L., H.W., X. Lian, C.Z., R.T., S.D., W.Z., P.L., X.S., Q.S., E.Z., and C.G. edited and revised the manuscript.

## Competing interests

The authors declare no competing interests.
