## [Transparent Peer Review file · Nature Communications]

Response to Reviewers' Comments

We greatly appreciate the opportunity to revise our manuscript and thank all anonymous reviewers for their constructive comments. The manuscript has been revised and improved according to the reviewers' valuable comments and suggestions. We hope that the revision will make it more acceptable for publication. Below are the point-by-point responses to the comments, along with the revision of the manuscript (typed in Italics and Arial) and the location of the revision. The line numbers referred to are for the clean version of the revised manuscript.

Reviewer #1 (Remarks to the Author):

The authors analyse the biophysical feedback of forestation. For this they use observational data, comparing the feedback for surface and air temperature. The main finding of the first part is that air temperature is less sensitive to tree cover changes than surface temperature. After establishing this, the authors analyse this difference in sensitivity by using Monxin Obukhov and quantify the maximum local climate effect. Finally, a comparison is made with the biochemical feedback of forestation, which shows that in specific regions the biophysical feedback can outperform the biochemical feedback.

The authors show an innovative approach of combining different data sources and analytical methods in order to be able to analyse the effect of forestation of air temperature using observational data. This is interesting, sinche such topics are usually tackled by using model data. I specifically enjoyed the analysis of the near surface temperature profiles and the decomposition of the air temperature sensitivity.

That being said, it took me quite some time to understand which dataset was used for which analysis and how the different data sources were combined and analysed. In my opinion, clarifying this aspect would be a big help in improving the paper. If the authors choose to rewrite/reorganize parts of the paper, some questions that I had during reading (some of which got resolved, but were still confusing in the beginning) could maybe serve as a guideline as to which parts to clarify:

Response: We appreciate the positive comments by the reviewer. We have carefully considered all the comments and suggestions and made corresponding point-by-point responses. In particular, we have revised the method section and clarified the usage of different datasets to improve the readability of the manuscript. Please see our detailed responses below.

1) Line 230: I thought we were only looking at the difference between 0 and 100% tree cover?

Response: Thanks for the comment. In the original manuscript, the section is based on the tree restoration potential (ΔTC), ranging from 0 to 100%. According to the comment of reviewer #3, we removed the analyses related to ΔTC and rewrote this section by comparing the biophysical and biochemical sensitivity of tree cover change from 0 to 100% (**Line 246**). This revision makes our results clearer that only local effects are evaluated and the non-local or global effects are not

included.

Comparison of biophysical with biochemical effects based on two temperature metrics

Most assessments of the climate benefits related to forestation have concentrated on the carbon sequestration (i.e., biochemical effect). Here, the biomass carbon stock sensitivity to tree cover is estimated by space-for-time analogy and converted to CO₂ absorption equivalents ($\delta\text{CO}_2e^{bchem}$) to represent the biochemical effect. We also convert the biophysical Ts and Ta sensitivities to the metric of equivalent CO₂ uptake ($\delta\text{CO}_2e^{bph,Ts}$ and $\delta\text{CO}_2e^{bph,Ta}$). These allow the comparison of the local biophysical and the biochemical climate effect and evaluate the relative importance of the former.

2) Line 399: where do these radiation values come from, which dataset?

Response: Thanks for raising this concern. Upward and downward longwave radiation data are from FLUXNET datasets. We have added the description of the data source (**Line 470**):

LW_u and LW_d represent upward and downward longwave radiation from the FLUXNET2015 dataset, respectively;

3) Figure 1: is this figure based on GLOBMAP? I can't find that anywhere.

Response: Thanks for the careful reading. Figure 1 is based on GLOBMAP dataset. We have added the description in the figure caption (**Line 129**):

The tree cover map for the sensitivity estimation is from the GLOBAMAP dataset.

4) Line 355: why this 10%? As in, why at all, and why exactly 10%?

Response: Thank you for raising this question. Here, we set this threshold for two reasons. First, according to the definition from the Food and Agricultural Organization (FAO), forests are defined as land with a tree crown cover larger than 10% and an area of more than 0.5 hectares (ha). Second, the temperature variations in low tree cover (e.g. <10%) pixels may be induced by the rest non-forest land cover components. Using these pixels for regression may reduce the reliability of estimated temperature sensitivity to tree cover changes. In the revised manuscript, we have added explanations of the threshold (**Line 415**):

Specifically, for each 0.25° × 0.25° grid cell, we filter out pixels with more than 1% water body coverage or less than 10% tree cover according to the forest definition by the Food and Agricultural Organization⁷⁹. This process is to reduce the impact of non-forest land cover types on the estimation of temperature sensitivity.

79. Keenan, R. J. et al. Dynamics of global forest area: Results from the FAO Global Forest Resources Assessment 2015. *Forest Ecology and Management* vol. 352 9–20 (2015).

5) The satellite based Ts and Ta are for 2010 whereas the potential tree cover map is for 2020 (lines 311–326). Is this difference in time what you correct for in lines 488–489? Unfortunately I don't really understand what you're trying

to say here.

Response: Thank you for raising this concern. In the original manuscript, we use tree cover data of 2010 to estimate sensitivities, considering the data availability of both GLOBMAP and GFC datasets. We also use tree cover data of 2020 from GLOBMAP to calculate tree restoration potential (ΔTC) under the current stage. According to the comment of reviewer #3, the analyses based on ΔTC are removed, and only tree cover data of 2010 are used in the revised manuscript.

We have rewritten the “Tree Cover Map” subsection to clarify this issue (**Line 377**):

In this study, two tree canopy cover datasets derived from different sensors are used for the analysis, including the recently released GLOBMAP fractional tree cover map with a spatial resolution of 250 m, and the Global Forest Change (GFC) tree cover map with a spatial resolution of 30 m. Considering the data availability, we use the tree cover maps of 2010 (TC_{2010}) of both two products to calculate the biophysical temperature sensitivities to ensure robustness. Both TC_{2010} maps are preprocessed and spatially aggregated to a 1 km resolution for further analysis (Supplementary Fig. 12).

Fig. S12. Global maps of fractional tree cover from (a) GLOBMAP and (b) Global Forest Change (GFC) datasets.

6) What I found most confusing is the different definitions/meanings/sources of T_s and T_a throughout the paper. When describing the satellite based T_s and T_a : „ T_s represent the mixture temperature of the tree canopy and the exposed soil at the observed angle “. So, T_s is at canopy height. Later is mentioned that T_a is located at screen height (line 339), what is screen height? Is that then above the canopy layer in the case of a forest? In line 430 and further, T_a is

normalized to 2m height. Why is this not done right away? But also, are we then looking at a T_a inside the canopy, (heavily influenced by the microclimate in a forest) or is it 2m above the canopy? Unfortunately, I also don't really understand formulas 1 and 2 and where they come from (see also a later comment on why a linear relationship is assumed) so I have difficulty assessing if all these differences in meaning are problematic or not (since you're comparing deltas and not actual values). And then in line 396, „ T_a is measured above the vegetation canopy, ranging from several decimeters to over ten meters “, so which height is actually used for T_a ? Clearing all this up would be extremely helpful.

Response: Thank you very much for these important issues. First, the evaluated T_a , in both remote sensing- and site-based analyses, indicates the air temperature of about 2 m above the land surface. Here, “land surface” indicates the interface layer between different land surface components and the atmosphere (e.g., forest canopy, rooftops and soil). Thus, for forested areas, T_a refers to the air temperature of about 2 m above the tree canopy. For clarity, we have revised the "screen height" to “approximately 2 m above the land surface”. We also added a schematic figure (**Supplementary Fig. 2**) and descriptions of the temperature metrics in the introduction section (**Line 78**):

Notably, the evaluated T_s indicates the radiometric temperature of the land surface, and T_a indicates the air temperature at 2 m above the land surface (Supplementary Fig. 2). The land surface here refers to the interface layer between different land components and the atmosphere (e.g. vegetation canopy)⁴⁴.

Fig. S2. Schematic representation of evaluated land surface temperature (T_s) and near surface air temperature (T_a).

Reference:

44. Reiners, P., Sobrino, J. & Kuenzer, C. Satellite-Derived Land Surface Temperature Dynamics in the Context of Global Change—A Review. *Remote Sens.* 15, (2023).

We also added descriptions of the air temperature in the method section (**Line 397**):

The other temperature metric used for assessment is T_a , the air temperature at

approximately 2 m above the interface layer between land components and the atmosphere. Specifically, for forested areas, the reference plane is the canopy, whereas for openlands the reference plane is approximately the ground (Supplementary Fig. 2).

Second, since T_a is measured above the vegetation canopy, ranging from several decimeters to over ten meters, we normalized FLUXNET T_a to explore the temperature profile from zero-plane displacement to the fixed 2-meter height above the canopy.

Third, about Equations 1 and 2, please see the later response to your critical comment on the assumption of a linear relationship.

Other comments:

Line 28: „different biophysical characteristics “ → different than what? I suggest changing to „several biophysical characteristics “, or something along those lines.

Response: Thanks for the suggestion. We have replaced the “different” with “several” (Line 36):

Meanwhile, forests present several biophysical (bph) characteristics, such as lower albedo and greater roughness length, resulting in the local cooling or warming effect compared to their neighboring openlands.

Line 32: what is a negative and what is a positive effect? Maybe use warming and cooling instead?

Response: Thanks for the comment. We have changed “negative/positive” to “cooling/warming” (Line 40):

The sign and magnitude of the local biophysical temperature effects can vary considerably based on spatial location and background climate, and are typically characterized by a shift from cooling effects in the tropics to warming effects in cold regions.

Line 67: what does b^{gp} actually stand for? (just out of curiosity, knowing this would help with reading)

Response: Thanks for the suggestion. In the original manuscript, the superscript “b^{gp}” is the abbreviation for “biogeophysical”. We have changed the superscript to “b^{ph}” as the abbreviation for biophysical for clarity (Line 36):

Meanwhile, forests present several biophysical (bph) characteristics, such as lower albedo and greater roughness length, resulting in the local cooling or warming effect compared to their neighboring openlands.

Line 85: only one other study is used in the figure for comparison?

Response: Yes, we compare our result with Duveiller et al, which provides biophysical maximum,

minimum and mean T_s effects of forestation. To avoid ambiguity, we have revised the sentence (Line 102):

The estimated δT_s^{bph} aligns well with a previous study of the potential T_s effect of forestation based on the unmixing method⁴⁶, suggesting the robustness to different analytical approaches.

Reference:

46. Duveiller, G., Hooker, J. & Cescatti, A. The mark of vegetation change on Earth's surface energy balance. *Nat. Commun.* 9, 64 – 75 (2018).

Line 88: valueS

Response: Thanks for the careful reading. We have corrected the mistake as the reviewer suggested (Line 105):

In terms of the magnitude, δT_a^{bph} demonstrates much lower absolute values compared to δT_s^{bph} .

Line 144-147: please rewrite this sentence, or maybe it already works if the first „While “ is removed?

Response: Thank you for the careful reading. We have rewritten the sentence and now it reads (Line 179):

However, we show that the ratios of T_a sensitivity slopes to T_s sensitivity slopes are close in the RS-based (16.9%) and site-based (17.4%) results (Fig. 2c and d). This result suggests that site measurements corroborate the relative magnitude of the RS-based maximum temperature sensitivity.

Figure 2: The text of the legend is very small, which makes it difficult to distinguish between T_s and T_a (the s and a look the same). If possible rearrange legend and panels to be able to increase font size (maybe the legend outside of the panel?)

Response: We thank the reviewer for the helpful suggestion. To better distinguish the two temperatures, we have changed T_s and T_a to T_s and T_a . We also rearranged the panels to increase the font size of the legend (Fig. 2).

Figure 2. ...

Subsection starting at line 163: What could be a valuable addition is a short discussion on the differences between the sites. Why is the influence of ra different between the three observed cases (Europe vs North America and Australia)?

Response: Thanks for the suggestion. We have added an explanation of the higher contribution from H in European winter than North American or Australian summer (**Line 226**):

We note that the contribution of H is greater in European winter than in North American or Australian summer. The possible reason is that H is more dominant in the turbulent flux exchange in winter (characterized by the higher Bowen ratio) than in summer⁵¹, thus contributing more to the temperature gradients between the land surface and the near-surface air, and further to the attenuation of the air temperature response.

Reference:

51. Lin, H., Li, Y. & Zhao, L. Partitioning of Sensible and Latent Heat Fluxes in Different Vegetation Types and Their Spatiotemporal Variations Based on 203 FLUXNET Sites. *J. Geophys. Res. Atmos.* 127, (2022).

Line 245: after how many years would this amount be reached? Compared to how fast trees grow and the biophysical effect will be apparent? Trees grow slow and biophysical and biochemical feedback act on different timescales, a sentences or two about this would be helpful I think.

Response: Thanks for the professional suggestion. We have added a sentence to discuss the temporal scale of two effects (**Line 320**):

Both evaluated biophysical and biochemical effects represent potential cumulative results. It may take a shorter period for biophysical processes (a single decade) to come into effect than biochemical processes, as the newly restored forests gradually absorb CO₂ and reach equilibrium after several decades.

Line 261: which regions?

Response: Thanks for the careful reading. Following reviewer #3's comment, this section about the comparison between biophysical and biochemical effects has been rewritten. Please see the response below for the new content.

Line 292: and A machine-learning model

Response: Thanks for pointing out this and the following mistakes. According to the comment of reviewer #3, the analyses using the machine-learning model for sensitivity extrapolation are removed. Correspondingly, the related descriptions and discussions are deleted.

Line 314: Both and two → remove one

Response: Revised as you suggested (**Line 382**):

Both TC₂₀₁₀ maps from GLOBMAP and GFC are preprocessed and spatially aggregated to the 1 km resolution for further analysis.

Line 318: by A Random Forest model

Response: Contents related to the random forest model are deleted for the same reason mentioned above.

Line 323: annual precipitation is mentioned twice

Response: This paragraph is deleted for the same reason mentioned above.

Line 357: water coverage AND elevation data

Response: Revised as you suggested (**Line 420**):

The water coverage and elevation data are from Joint Research Center Global Surface Water Mapping Layers v1.4 and GMTED2010 datasets, respectively.

Lines 360–363: why do you assume the relation is linear? Do you have a reference for this (pretty strong) assumption? And about the formulas: what is the difference between TC^T and TC ? Also, ΔT_{sbgp} and ΔT_{abgp} are for 100% increase in forest cover (I thought?), how is this represented in formula 1 and 2?

Response: Thanks for raising these concerns. The non-linear relationship is generally discussed in those studies at finer scales (Zhao et al., 2023) or inner-city scales (Yang et al., 2024). For most large-scale or global studies, the linear assumption is widely accepted and used (Alkama and Cescatti, 2016; Wang et al., 2023; Zhang et al., 2024). We add these references for the linear

relationship between tree cover and temperature (**Line 423**):

After the screening process, δT_s^{bph} and δT_a^{bph} can be estimated using a linear regression model between tree cover and corresponding temperature values for each 0.25° grid (Alkama and Cescatti, 2016; Wang et al., 2023; Zhang et al., 2024).

Second, the original formulas 1 and 2 are the matrix representation of the ordinary least squares (OLS) method for estimating the slope, and the superscript “T” denotes the matrix transpose. We rewrite the formula 1 and 2 as the form of linear relationship for clarity, and it now reads (**Line 425**):

$$T_s = \delta T_s^{bph} \times TC_{2010} + b_s \quad (1)$$

$$T_a = \delta T_a^{bph} \times TC_{2010} + b_a \quad (2)$$

where, b_s and b_a are the regression intercepts.

References:

- Alkama, R., Cescatti, A., 2016. Biophysical climate impacts of recent changes in global forest cover. *Science* (80-.). 351, 600 – 604. <https://doi.org/10.1126/science.aac8083>
- Wang, H., Yue, C., Luysaert, S., 2023. Reconciling different approaches to quantifying land surface temperature impacts of afforestation using satellite observations. *Biogeosciences* 20, 75 – 92. <https://doi.org/10.5194/bg-20-75-2023>
- Yang, L., Ge, J., Cao, Y., Liu, Y., Luo, X., Wang, S., Guo, W., 2024. Enhanced Cooling Efficiency of Urban Trees on Hotter Summer Days in 70 Cities of China. *Adv. Atmos. Sci.* 41, 2259 – 2275. <https://doi.org/10.1007/s00376-024-3269-9>
- Zhang, Y., Wang, X., Lian, X., Li, S., Li, Y., Chen, C., Piao, S., 2024. Asymmetric impacts of forest gain and loss on tropical land surface temperature. *Nat. Geosci.* 13, 823 – 831. <https://doi.org/10.1038/s41561-024-01423-3>
- Zhao, J., Zhao, X., Wu, D., Meili, N., Fatichi, S., 2023. Satellite - based evidence highlights a considerable increase of urban tree cooling benefits from 2000 to 2015. *Glob. Chang. Biol.* <https://doi.org/10.1111/gcb.16667>

Line 365: sample size of what?

Response: Thanks for the comment. The sample size indicates the number of samples for the linear regression model. The sentence now reads (**Line 427**):

To ensure the reliability of the results, biophysical sensitivity calculation is performed only when the total sample size of the linear regression model exceeds 90 (more than 10% of pixels within the 0.25° grid) and the difference between the highest and lowest tree cover is greater than 40%.

Line 382: this is not entirely true though, is it? Because of the spatial autocorrelation and the fact that a forest influences its surroundings? Removing the forest could also change the temperature on the non-forest site.

Response: Thanks for the valuable comment. We admit that forestation can affect the temperature of the surrounding area, by affecting heat advection or the circulation pattern. We have clarified that

the paired site method can only be used to validate the “local” temperature effect derived from remote sensing data (Line 450):

The RS-based local temperature sensitivity can be validated through the differences of measurements between spatially adjacent paired forest and non-forest sites.

Line 401: why is the lack of spatial pairing a reason for this? I understood from the previous paragraph that no spatial pairing approach is used? As I understand from the formula, you simply average over all forest/open land sites anyway?

Response: Thanks for pointing this out. The spatial pairing approach is not used here and we have removed this redundant description (Line 472):

We bin the SW_d with a fixed interval of 10 w/m^2 and calculate the mean δT_s^{bph} or δT_a^{bph*} to explore the sensitivity variability under different background radiation conditions.*

Second, we average the forest and openland observations in different shortwave radiation bins, rather than all possible observations. We added a flow chart for the methodology for clarity (Supplementary Fig. 8).

Fig. S8. Flow chart of the remote sensing (RS)-based local temperature effect validation

using FLUXNET and gridded temperature data.

Reviewer #2 (Remarks to the Author):

Content of the paper

In their paper, "The Overlooked Local Air Warming Mitigation of Potential Tree Restoration", the authors provide insights on the biophysical impacts of forestation. Namely, they compute the impacts of forestation on air temperature (as opposed to surface temperature) and compare these impacts with the often reported surface temperature changes, and with biogeochemical effects.

For this, the authors train an RF model: From meteorological, soil, forest cover, and topographic data, this predicts the maximum forest cover under current climatic conditions across the globe.

They correlate weather station air temperature T_a to satellite-observed surface temperature T_s , to get global coverage of T_a . Then, after a filtering process to account for too-high elevation differences or too-low forest cover differences within that pixel, linear relationships between tree cover and the two temperatures are created,

$$T_s = TC_{2010} \times dT_s$$

$$T_a = TC_{2010} \times dT_a$$

From this relationship, they can estimate the temperatures within that pixel if it contained the maximum possible forest cover. The authors validate their approach with site data. They finally compare their results with biogeochemical impacts of forestation.

Remarks

In general, I find this to be a relevant and well-crafted paper that could be suitable for publication within the journal Nature Communications. It offers substantial new contributions. The authors did a good job in using various data sources to back their claims and I enjoyed reading this paper.

However, some more work needs to be invested to make the paper and its contribution clearer which is why I categorize this manuscript as needing "major revisions".

For instance, it needs to be highlighted more what the new results are compared to existing knowledge, for instance compared to the study by Windisch et al. (2019), which the authors also cite. I found the state of current knowledge to be addressed, but not clear enough. Perhaps (only an idea), some more consolidation of current state of knowledge would be helpful, and the main results of this new manuscript should be made more prominent.

Many points throughout the manuscript can be made clearer, it is sometimes not clear whether local impacts, or global impacts are talked about. And in various points (see my detailed comments below) the explanations need some more clarity. In the assessment of biogeochemical impacts, soil is neglected.

Please find my detailed comments on the various sections below.

Response: We appreciate your positive comments. We have revised our manuscript carefully and the detailed corrections are listed below. Specifically, we have invested more work to clarify the

new contribution of this paper compared to previous Ts-based studies (e.g. Windisch, et al.). The state of current knowledge, such as global/local effects, the influence of soil carbon, and the widespread impact of forestation, have been introduced or discussed. We also revised explanations and technical terms to make the paper clearer. We hope the revised paper will be more suitable for publication.

Abstract

1. 12-13: flatter temp profiles in forested areas *compared to non-forested*

Response: Thanks for the comment. The sentence has been revised as suggested (**Line 14**):

We further attribute the discrepancy in temperature responses to the reduced aerodynamic resistance and the resultant flatter near-surface temperature profiles in forests compared to non-forests.

1. 14 the substantial climate benefits, could mention here how high they will get, how many K? How much is it compared to surface temperature? Also, what is the potential global impact?

Response: Thanks for the suggestion. According to the comment of reviewer #3, we have updated the results of the comparison between biophysical and biochemical effects. The new results focus on the local biophysical and biochemical sensitivities, rather than the global biophysical and biochemical climate effects of potential tree restoration. Similar to Windisch, et al., the current comparisons are based on the equivalent CO₂ uptake, rather than the temperature effect (K). This revision makes our results clearer that only local effects are evaluated and the non-local or global effects are not included. Correspondingly, we have rewritten the abstract (**Line 4**):

Forestation, as a key component of nature-based solutions, has the potential to cool or warm the local climate through biophysical processes, thereby enhancing or offsetting the global warming mitigation from carbon assimilation. Currently, the magnitude of such biophysical effects on local climate remains unconstrained, as most previous observational studies rely on land surface temperature (Ts), rather than the policy-relevant near surface air temperature (Ta). Based on satellite observations, we show that the Ta response to tree cover change is significantly lower, ranging from 15% to 30%, compared to Ts response. The relative magnitude of the two temperature effects is supported by independent evidence from site observations. We further attribute the discrepancy in temperature responses to the reduced aerodynamic resistance and the resultant flatter near-surface temperature profiles in forests compared to non-forests. Moreover, we show that at mid- or northern high-latitudes, the maximum seasonal biophysical Ta warming or cooling only accounts for approximately 10% of the equivalent climate effect of carbon sequestration in terms of magnitude, whereas the ratio for the biophysical Ts effect can reach 40%. These results highlight that Ts-based assessments may significantly overestimate the local climate impact of tree cover change. We also emphasize that when evaluating the biophysical effects of forestation or deforestation, the proper temperature metric should be specified and used in different cases to avoid misleading conclusions.

l. 18: but the key message is not the strong potential, but quite contrarily, the much lower potential than what we anticipate, I think the wording needs to be a bit different **(Line 14)**:

Response: Thanks for the constructive comment. We've revised the wording according to the new results:

Moreover, we show that at mid- or northern high-latitudes, the maximum seasonal biophysical Ta warming or cooling only accounts for approximately 10% of the equivalent climate effect of carbon sequestration in terms of magnitude, whereas the ratio for the biophysical Ts effect can reach 40%. These results highlight that Ts-based assessments may significantly overestimate the local climate impact of tree cover change. We also emphasize that when evaluating the biophysical effects of forestation or deforestation, the proper temperature metric should be specified and used in different cases to avoid misleading conclusions.

Introduction

l. 26 - reference 5: there is a new version of this paper by the same authors: <https://doi.org/10.1038/s41586-024-07602-x>

Response: Thanks for the careful reading. We have updated the reference **(Line 598)**:

Reference:

5. Pan, Y. et al. The enduring world forest carbon sink. *Nature* **631**, 563–569 (2024).

l. 39 - some more introduction on local, non-local, and global effects would be nice. There is quite some work from the group of Julia Pongratz on this topic.

Response: Thanks for the constructive comment. We added an introduction about non-local and global effects **(Line 40)**:

Forest changes also affect the temperature of spatial nearby regions through advective transport, and even global temperature via altering the large-scale circulation patterns¹⁴. The magnitude of this non-local effect depends on the area extent and the geolocation of the changes^{15,16}.

Reference:

14. Pongratz, J. et al. Land Use Effects on Climate: Current State, Recent Progress, and Emerging Topics. *Curr. Clim. Chang. Reports* **7**, 99 - 120 (2021).

15. Winckler, J., Lejeune, Q., Reick, C. H. & Pongratz, J. Nonlocal Effects Dominate the Global Mean Surface Temperature Response to the Biogeophysical Effects of Deforestation. *Geophys. Res. Lett.* **46**, 745 - 755 (2019).

16. Winckler, J., Reick, C. H. & Pongratz, J. Why does the locally induced temperature response to land cover change differ across scenarios? *Geophys. Res. Lett.* **44**, 3833 - 3840 (2017).

l. 54 - which limitations?

Response: Thanks for the comment. We have revised the sentences to clarify the limitation **(Line**

63):

According to the report of the Intergovernmental Panel on Climate Change (IPCC), the indicator used to describe global land warming and frame climate change mitigation targets is near surface air temperature (Ta) rather than Ts³⁶. Despite the strong correlation between Ts and Ta³⁷, the Ts effect of forest change may significantly differ from the Ta effect³⁸. Ts-based assessments are useful for model refinement or informing the sign of Ta effects, but the values cannot be directly considered in climate treaties or policies.

Reference:

36. *Intergovernmental Panel on Climate Change. Changing State of the Climate System. in Climate Change 2021 – The Physical Science Basis 287 – 422 (Cambridge University Press, 2023). doi:10.1017/9781009157896.004.*
37. *Wang, Y. R., Hessen, D. O., Samset, B. H. & Stordal, F. Evaluating global and regional land warming trends in the past decades with both MODIS and ERA5-Land land surface temperature data. Remote Sens. Environ. 280, 113181 (2022).*
38. *Mildrexler, D. J., Zhao, M. & Running, S. W. A global comparison between station air temperatures and MODIS land surface temperatures reveals the cooling role of forests. J. Geophys. Res. Biogeosciences 116, 1–15 (2011).*

1. 57 – Note that the IPCC actually always talks about LSAT, land surface air temperature, and not surface temperature.

Response: Thanks for pointing this out. We have highlighted the temperature used in IPCC reports (Line 64):

According to the report of the Intergovernmental Panel on Climate Change (IPCC), the indicator used to describe global land warming and frame climate change mitigation targets is land surface air temperature (Ta) rather than Ts³⁶.

Reference:

36. *Intergovernmental Panel on Climate Change. Changing State of the Climate System. in Climate Change 2021 – The Physical Science Basis 287 – 422 (Cambridge University Press, 2023). doi:10.1017/9781009157896.004.*

1. 58 – citation 34: there is a new report, also the citation seems wrong? at least the one I find has a different citation: https://www.ipcc.ch/site/assets/uploads/2017/09/WG1AR5_Chapter02_FINAL.pdf

Response: We thank the reviewer for the careful reading. We have updated the reference (Line 665):

Reference:

36. *Intergovernmental Panel on Climate Change. Changing State of the Climate System. in Climate Change 2021 – The Physical Science Basis 287 – 422 (Cambridge University Press, 2023). doi:10.1017/9781009157896.004.*

l. 60 you criticize the sparse distribution of study sites in other studies, are the used study sites better distributed?

Response: Thanks for this critical comment. The distribution of paired sites is limited in Europe and North America (Chen et al., 2018). Here, the proposed methodology makes full use of more than 100 sites spread across five continents.

Reference:

Chen, L., Dirmeyer, P.A., Guo, Z., Schultz, N.M., 2018. Pairing FLUXNET sites to validate model representations of land-use/land-cover change. *Hydrol. Earth Syst. Sci.* 22, 111–125. <https://doi.org/10.5194/hess-22-111-2018>

We have highlighted the limitation of the current paired site in the revised manuscript (**Line 68**):

Although a few studies have explored the different responses of these two temperatures in the context of forest change, their results may be affected by the uncertainties in numerical models or the sparse distribution of paired forest and non-forest sites.

l. 67 = 100% increase in tree cover, what does that mean? What is the spatial moving window approach in that regard?

Response: Thanks for the comments. First, the hypothetical tree cover gain from 0 to 100% indicates an ideal forest change. The evaluated δT_s^{bph} or δT_a^{bph} is based on this ideal change regardless of the existing tree cover. Second, according to your comment below, we used the “spatial-for-time analogy” to replace “the spatial moving window” for clarity (**Line 76**). We also added a schematic figure to show how the methodology works (**Supplementary Fig. 1**).

We first estimated the biophysical T_s and T_a sensitivity to a hypothetical 100% increase in tree cover (denoted as δT_s^{bph} and δT_a^{bph}) at the 0.25° scale, based on the space-for-time analogy (Supplementary Fig. 1)

Fig. S1. Schematic representation of the methodology for estimating land surface temperature (T_s) or air temperature (T_a) sensitivity to tree cover change (δT_s^{bph} and δT_a^{bph}). (a–f) The example grid (59.75°–60°N, 63.5°–63.75°E) with positive δT_s^{bph} and δT_a^{bph} in January 2010. (a) True color image of the 0.25° grid. (b) GLOBMAP 2010 tree cover map. (c) Mean T_s of January 2010. (d) Mean T_a of January 2010. (e) Linear regression for estimating positive δT_s^{bph} of the grid. (f) Linear regression for estimating positive δT_a^{bph} of the grid. (g–l) Similar to (a–f), but for the other example grid (44.75°–45°N, 38°–38.25°E) with negative sensitivities in July 2010.

l. 67 not a big deal, but kind of strange that it's always "biophysical" but the abbreviation is "bpg" which I guess stands for biogeophysical

Response: Thank you for pointing this out. We have changed the superscript to "bph" as the abbreviation for biophysical for clarity.

l. 70 – globally distributed – you mean FLUXNET, right? Just mention this here. Globally distributed suggests that they are somewhat uniformly distributed across the globe, but Fluxnet has big biases towards certain regions (East Asia, North America, Europe).

Response: Following your suggestion, we have revised the data statements (**Line 84**):

Furthermore, we use the FLUXNET2015 dataset⁴⁵ and two gridded temperature

datasets to validate the differences between two sensitivities and elucidate the underlying biophysical mechanisms.

Reference:

45. Pastorello, G. et al. *The FLUXNET2015 dataset and the ONEFlux processing pipeline for eddy covariance data. Sci. Data 7, 1 – 27 (2020).*

l. 72 – spatially gap fill: you mean to extrapolate to the globe?

Response: Thanks for the comment. According to the suggestion of reviewer #3, the analyses based on sensitivity extrapolation are removed. The related descriptions and results are deleted.

also not a big deal, but your use of tenses is inconsistent. Sometimes present, sometimes past.

Response: Thanks for the careful reading. We have unified the tenses into the present tense.

I am missing a clear definition of T_a which is, I assume, 2m above ground.

Response: Thanks for raising this important question. The evaluated T_a is not exactly the air temperature 2 m above the ground, but the air temperature above the land surface (or underlying surface) (Zhang et al., 2022). Here, land surface indicates the interface layer between different land surface components and the atmosphere (e.g., tree canopy, rooftops and soil) (Reiners et al., 2023). Thus, for forested areas, T_a refers to the air temperature of about 2 m above the tree canopy; while for openlands, T_a is approximately air temperature 2 m above the ground.

References:

Reiners, P., Sobrino, J., Kuenzer, C., 2023. Satellite-Derived Land Surface Temperature Dynamics in the Context of Global Change — A Review. *Remote Sens.* 15. <https://doi.org/10.3390/rs15071857>

Zhang, T., Zhou, Y., Zhao, K., Zhu, Z., Chen, G., Hu, J., Wang, L., 2022. A global dataset of daily maximum and minimum near-surface air temperature at 1km resolution over land (2003-2020). *Earth Syst. Sci. Data* 14, 5637 – 5649. <https://doi.org/10.5194/essd-14-5637-2022>

In the revised manuscript, we gave the definition of T_a for clarity (**Line 78**):

Notably, the evaluated T_s indicates the radiometric temperature of the land surface, and T_a indicates the air temperature at 2 m above the land surface (Supplementary Fig. 2). The land surface here refers to the interface layer between different land components and the atmosphere (e.g. vegetation canopy).

Fig. S2. Schematic representation of evaluated land surface temperature (T_s) and near surface air temperature (T_a).

We also added descriptions of the air temperature in the method section (Line 397):

The other temperature metric used for assessment is T_a , the air temperature at approximately 2 m above the interface layer between land components and the atmosphere. Specifically, for forested areas, the reference plane is the canopy, whereas for openlands the reference plane is approximately the ground.

Results

1. 87 very nice, this comparison with the Duveiller paper.

Response: Thanks for the positive comment.

1. 89 why is it called the "local" effect here.

Response: Thanks for the comment. We emphasize the effects "local" to distinguish them from non-local effects of tree restoration, which are not included in our RS-based analysis. We have rearranged the sentence for clarity (Line 105):

In terms of the magnitude, δT_a^{bph} demonstrates much lower absolute values compared to δT_s^{bph} ($-0.14 \text{ K} \pm 0.40 \text{ K}$ vs. $-0.65 \pm 1.22 \text{ K}$, global mean \pm standard deviation), indicating that the local T_a effect of tree restoration is approximately 22% of the T_s effect.

1. 92 I am confused, should it be $+0.17\text{K}$ vs $+0.53\text{K}$?

Response: Thanks for the careful reading. We have corrected the typo (Line 110):

At northern high-latitudes, the T_a -based warming induced by tree cover change is 32% of the T_s -based warming (0.17 K vs. 0.53 K).

1. 95 nice that you did this with multiple data products

Response: Thank you for recognizing our work.

1. 116 I like that you did this analysis for min and max

Response: Thanks for the positive comment.

1. 119 – I find it notable that for the max temp, even in boreal regions there is always cooling. This is important for applications / policy I believe

Response: Thank you for pointing this out. We have emphasized this result in the discussion section **(Line 341)**:

In particular, the diurnal and seasonal changes in biophysical temperature effects should be considered when formulating comprehensive forest-based policies. For instance, we observe maximum temperature cooling and minimum temperature warming effects of forests at high-latitudes. This suggests that tree restoration in such cold regions may be a solution to reduce the risks or impacts of daytime warming on the ecosystem.

1. 132 – why is this called the “ideal” temperature effect? Aren’t you just doing the same exercise but now for observational data and not RS data?

Response: We call the effect “ideal” because it is inferred from the observations from forest and non-forest sites, and the land cover changes do not actually occur. To avoid ambiguity, we have revised the sentence **(Line 149)**:

To ensure the robustness of our findings, especially the relative magnitude of T_a effects to T_s effects, we further validate the RS-based δT_s^{bph} and δT_a^{bph} against the temperature effects of forestation (δT_s^{bph} and δT_a^{bph*}) inferred from the in-situ observations and gridded temperature data.*

Second, the validation based on FLUXNET observations is not exactly analogous to the method used for RS data. We provided a figure to illustrate how it works **(Supplementary Fig. 8)**, which consists of the following steps:

- (1) We first subtract elevation-corrected gridded temperature data from FLUXNET T_s or T_a . Here, the gridded data mainly reflect the macro-climate; FLUXNET T_s or T_a are driven by both the macro-climate and the land use. Thus, their difference represent the relative impact of land cover on local T_s or T_a (Novick and Barnes, 2023).
- (2) We then calculate the difference between corrected FLUXNET T_s or T_a in forest sites and non-forest sites (x_f and x_o in Fig. S8). The difference values represent the impact of potential forestation on T_s or T_a (non-forest converting to forest). Notably, we calculate this difference at each 10 w/m^2 shortwave radiation (SW_d) bin to represent the temperature effect of forestation under changing radiative conditions.
- (3) We estimate the relationship between shortwave radiation and the T_s or T_a effects from both RS and in-situ data (e.g. Fig 2a and 2b). We compare the regression results to validate (a) whether the variation in RS-based T_s and T_a sensitivity with radiation are quantitatively aligned with the

in-situ results. (b) whether the relative magnitude of RS-based Ta effects to Ts effects is supported by in-situ results.

Fig. S8. Flow chart of the remote sensing (RS)-based local temperature effect validation using FLUXNET and gridded temperature data.

Reference:

Novick, K.A., Barnes, M.L., 2023. A practical exploration of land cover impacts on surface and air temperature when they are most consequential. *Environ. Res. Clim.* 2, 025007. <https://doi.org/10.1088/2752-5295/acdf9>

1. 134 gradient?

Response: Thanks for the careful reading. We have replaced “gradient” by “bins” for clarity (**Line 153**):

Here, δT_s^{bph} and δT_a^{bph*} are estimated in different shortwave radiation (SW_d) bins to represent the relative changes with changing background radiation conditions (Methods, Supplementary Fig. 8).*

Fig 2b) provide n, and indicate that this is from fluxnet data. Is a line plot

ok here? I think you should plot the actual data points, FLUXNET should be around 100-200 data points if I am not mistaken.

Response: Thanks for the suggestions. First, we noted that the date is from FLUXNET in the figure caption (**Line 166**).

(b) *FLUXNET-based relationships between the mean temperature sensitivities (δT_s^{bph*} and δT_a^{bph*}) and SW_d .*

Second, the line plots in Fig.2b show the estimated δT_s^{bph*} or δT_a^{bph*} values rather than the original FLUXNET data. The forest and openland samples used to estimate each data point of the line plot have different sample sizes n . Hence, it is difficult to label n in the figure. For clarity, we here provide raw data (including forest n and openland n for each data point) for Figures 2b and 2d. The file, named “Raw_data_for_Fig2.xlsx”, will be published with the article if the manuscript can be accepted.

Why is it only mean and max, and not min?

Response: Thank you for the comment. We have added minimum temperature results (**Line 184**):

We also perform similar analyses to the site-based minimum T_a and T_s sensitivities ($\delta T_{s_{min}}^{bph}$ and $\delta T_{a_{min}}^{bph*}$, Supplementary Fig. 10), which supports the lower T_a -based warming than T_s -based warming during the nighttime in the RS-based results. We note that the relationship between $\delta T_{a_{min}}^{bph*}$ and SW_d is not significant, which corresponds to the weak correlation between $\delta T_{a_{min}}^{bph}$ and SW_d in the RS-based results ($r=-0.38$).*

Fig. S10. Validation of the monthly minimum land surface temperature and air temperature sensitivities. (a) Remote sensing-based relationships between minimum temperature sensitivities ($\delta T_{s_{min}}^{bph}$ and $\delta T_{a_{min}}^{bph}$) and background shortwave radiation (SW_d). (b) FLUXNET-based relationships between minimum temperature sensitivities ($\delta T_{s_{min}}^{bph*}$ and $\delta T_{a_{min}}^{bph*}$) with SW_d , using Climatic Research Unit (CRU) temperature data to exclude the impact of macro-climate background. (c) Same as (b), but the Berkeley Earth Surface Temperatures (BEST) data are used to exclude the impact of macro-climate background.

The usage of different y scales is quite misleading here. This should be changed.

Response: Thank you for this comment. We have unified the y scales (Fig. 2).

Figure 2. ...

Is this now monthly mean and max?

Response: Thank you for this comment. Yes, the data for validation are at the monthly scale. We have declared the temporal scale in the figure caption (Line 164):

Figure 2. Validation of the monthly land surface temperature and air temperature sensitivities.

Why not a figure like 1c for the sites? This would be a convincing comparison between RS and site data.

Response: Thank you for the suggestion about the visualization. Here, the FLUXNET-based sensitivities ($\delta T_s^{\text{bph}*}$ and $\delta T_a^{\text{bph}*}$) are calculated at different radiation bins and thus do not represent the results of specific geo-locations or latitudes. Thus, these results are unlikely to be presented in the manner of Fig. 1c.

l. 143 thank you for providing this disclaimer. can you also provide what the revisit times are? Is it from the MODIS with 4 visits per day? And here you mean occurrence of _daily_ maximum temperature?

Response: Thank you for the comments. We have added the overpass time and the word “daily” as you suggested (Line 176):

The slopes derived from in-situ measurements are more pronounced than RS-based results, which may be due to the satellite overpass times (about 13:30, see Methods) not precisely coinciding with the occurrence of daily maximum temperatures.

We also provide four overpass times of MODIS Ts in the method section (Line 390):

Here, the monthly mean Ts data of 2010 is generated by the daily four observations from Moderate Resolution Imaging Spectroradiometer (MODIS) onboard Terra and Aqua satellites (observed at 1:30, 10:30, 13:30 and 22:30).

Section "Biophysical mechanisms..." is a very nice section but needs some more work to clear things up.

Fig 3 - very interesting figure, unfortunately rather low n, but I guess there's nothing to do about that. I however still do not understand why it is now about 2m above the canopy. Should it not be 2m above the ground that is relevant?

Response: Thanks for raising the concern. The evaluated Ta indicates the air temperature above the tree canopy in the forested area. Please refer to the response above about the definition of Ta.

We admit that the air temperature 2 m above ground (usually below the tree canopy) is relevant for many organisms or processes within forests. However, the evaluated Ta is also important, because (a) Ta is close to the definition of 'Tas' in climate models (the inferred temperature at 2 m above zero-plane displacement height), which is used to assess climate change (Winckler et al., 2019). (b) Ta is more relevant to boundary layer dynamics, rainfall initiation, and interaction between the climate system and land surface (Novick and Katul, 2020). Therefore, we believe that the evaluated Ta is a highly relevant indicator for climate change and land-atmosphere interaction.

References:

- Winckler, J., Reick, C.H., Luyssaert, S., Cescatti, A., Stoy, P.C., Lejeune, Q., Raddatz, T., Chlond, A., Heidkamp, M., Pongratz, J., 2019. Different response of surface temperature and air temperature to deforestation in climate models. *Earth Syst. Dyn.* 10, 473 - 484. <https://doi.org/10.5194/esd-10-473-2019>
- Novick, K.A., Katul, G.G., 2020. The Duality of Reforestation Impacts on Surface and Air Temperature. *J. Geophys. Res. Biogeosciences* 125, 1 - 15. <https://doi.org/10.1029/2019JG005543>

c) what is the Sum? The sum of all components? Why is it 0 then? Shouldn't it be the difference between 1.97 and 0.16K?

Response: Thank you for pointing this out. In the original manuscript, "Sum" means the simulated $\delta T_a^{\text{bph*}}$, represented by the sum of $\delta T_s^{\text{bph*}}$, δT^{ra} and δT^{H} . The estimated value is close to but not exactly zero. Following your suggestion, we have replaced the bar "Sum" with the difference between $\delta T_a^{\text{bph*}}$ and $\delta T_s^{\text{bph*}}$ ("Diff").

Figure 3. ... (c) Bar plots of the mean air temperature sensitivity (δT_a^{bph*}), land surface temperature sensitivity (δT_s^{bph*}) and their difference (Diff) contributed by variations in aerodynamic resistance (δT^{ra}), and sensible heat (δT^H)...

1. 169 but the 1.97 is so much higher than the reported 0.53 reported above?

Response: Thanks for your comment. The most likely reason for this difference is that the reported 0.53 K indicates the annual warming in high-latitude regions, and 1.97 K (1.63 K in the revised version) here indicates the winter warming in Europe. In addition, differences in the observation scales may contribute to the mismatches between RS-based and site-based results.

1. 173 – I am confused. Fig. 3a shows that the air temperature of forests is the same as the surface temperature for forests. Why then is the impact of forestation higher for surface temp compared to air temp (as was shown before)?

Response: Thanks for your comment. The main reason is that the temperature impact of forestation relies on both status in non-forest openland (before forestation) and forest (after forestation). Although the vertical variation in the forest temperature profile is less pronounced, we find significant temperature gradients between T_s and T_a in non-forest openlands. This gradient weakens the strong surface warming effect in Europe (Fig. 3a) or the strong summer cooling in North America and Australia (Fig. 3d and 3g).

1. 179 – steeper in openlands? In the figs, it is steeper in forests I would say?

Response: Thank you for the valuable comment. We have revised the word “steeper” to avoid ambiguity (**Line 208**):

These factors collectively result in more significant temperature gradients in openlands, ...

1. 216 – where do the $-0.03K$ come from? More specifically, why is this not the

-0.14K as reported in the first section of the results in l. 88? Is that the global average now? How does it compare to global average change in T_s ? This should be a more prominent result.

Response: Thanks for your comment. According to the comment of reviewer #3, the biophysical and biochemical climate effect analyses of the potential tree restoration are removed. We have rewritten this section by comparing the concept of sensitivity (**Line 245**). This revision makes our results clearer that only local effects are evaluated and the non-local or global effects are not included. Similar to Windisch, et al., the current comparisons are based on the equivalent CO_2 uptake, rather than the temperature effect (K). The new results allow us to investigate the relative importance of the biophysical effect and reveal the overestimation of the biophysical climate effect using T_s (rather than the more relevant T_a) as the metric.

Comparison of biophysical with biochemical effects based on two temperature metrics

Most assessments of the climate benefits related to forestation have concentrated on the carbon sequestration (i.e., biochemical effects)^{52,53}. Here, the biomass carbon stock sensitivity to tree cover is estimated via space-for-time analogy and converted to CO_2 absorption equivalents ($\delta CO_2 e^{bchem}$) to represent the biochemical effect. We also convert the biophysical T_s and T_a sensitivities to the metric of equivalent CO_2 uptake ($\delta CO_2 e^{bph,Ts}$ and $\delta CO_2 e^{bph,Ta}$, Supplementary Fig. 11). These allow the comparison of the local biophysical and biochemical climate effects and evaluation of the relative importance of the former.

The spatial map shows that $\delta CO_2 e^{bchem}$ in tropical rainforests can exceed 600 t/ha, which is much greater than temperate and boreal forests (Fig. 4a). This suggests that restoring damaged forests in tropical regions has the most carbon benefit. Latitudinally, $\delta CO_2 e^{bchem}$ at low-latitudes is higher than that at mid- or high-latitudes, with global mean of 268.2 ± 37.8 t/ha (mean \pm uncertainty) (Fig. 4b). In terms of the biophysical effect, $\delta CO_2 e^{bph,Ts}$ (41.7 ± 9.3 t/ha) provides a global average of 15.7% additional benefits to $\delta CO_2 e^{bchem}$ (Fig. 4b). However, if the more relevant biophysical T_a effect is considered, the ratio of $\delta CO_2 e^{bph,Ta}$ (9.3 ± 2.9 t/ha) to $\delta CO_2 e^{bchem}$ is only 3.5%.

We then focus on northern high-latitudes, where tree restoration shows a biophysical warming effect. The resultant negative $\delta CO_2 e^{bph,Ts}$ could offset 9.5% of the $\delta CO_2 e^{bchem}$ annually (Fig. 4b). The high-latitude biophysical warming is more pronounced in the cold season and can reduce the biochemical climate effect by 42.4% in March (Fig. 4c). However, when $\delta CO_2 e^{bph,Ta}$ is used as the indicator, the offset of biophysical to biochemical effects is only 3.3% at the annual scale, with the maximum monthly value of 10.6% (February) (Fig. 4b and 4c). In mid-latitudes, the seasonal $\delta CO_2 e^{bph,Ts}$ can enhance $\delta CO_2 e^{bchem}$ by up to 33.7% (northern hemisphere) and 40.5% (southern hemisphere) during summer. However, these seasonal ratios are only about 10% considering $\delta CO_2 e^{bph,Ta}$ (Fig. 4d and 4f). In low-latitudes, annual positive $\delta CO_2 e^{bph,Ts}$ is equivalent to 25.5% of $\delta CO_2 e^{bchem}$, while the ratio for $\delta CO_2 e^{bph,Ta}$ is

only 6.2%, with insignificant seasonal variations (Fig. 4b and 4e). These results suggest that the relative importance of biophysical effects largely depends on the evaluated temperature metric, and the role of biophysical effects in the overall climate effect (usually measured by T_a) may not be as important as estimated in previous T_s -based studies^{22,23}.

Figure 4. Comparison of the biophysical (bph) and biochemical (bchem) effects of potential tree cover gain. (a) Global pattern of the biochemical effect of potential tree cover gain (δCO_2e^{bchem}). (b) Global and latitudinal means of biochemical and biophysical effects of potential tree cover gain. The T_s -based and T_a -based biophysical effects are shown as the equivalent CO_2 uptake ($\delta CO_2e^{bph,Ts}$ and $\delta CO_2e^{bph,Ta}$). The error bars indicate the uncertainty of the mean. (c–f) Monthly ratios of T_a -based and T_s -based biophysical effects to equivalent biochemical effects across northern high-latitudes (>50°N), northern mid-latitudes (20°–50°N), tropics (20°S–20°N) and southern mid-latitudes (> 20°S). The shaded area indicates the uncertainty of the ratios.

References:

22. Windisch, M. G., Davin, E. L. & Seneviratne, S. I. Prioritizing forestation based on biogeochemical and local biogeophysical impacts. *Nat. Clim. Chang.* **11**, 867–871 (2021).
23. Zhu, L. et al. Comparable biophysical and biogeochemical feedbacks on warming from tropical moist forest degradation. *Nat. Geosci.* **16**, 244–249 (2023).
52. Walker, W. S. et al. The global potential for increased storage of carbon on land. *Proc. Natl. Acad. Sci. U. S. A.* **119**, 1–12 (2022).
53. Lewis, S. L., Wheeler, C. E., Mitchard, E. T. A. & Koch, A. Restoring natural forests is the best way to remove atmospheric carbon. *Nature* **568**, 25–28 (2019).

1. 214 – this gap-filling has not yet been explained, where do the gaps come from?

Response: Thanks for your comment. We have rewritten this section. Please refer to the response above for the revision.

l. 224 – unclear, “reverses to be negative”?

Response: Thanks for your comment. We have rewritten this section. Please refer to the response above for the revision.

Fig 4. e) instead of the ratio, it would be more understandable if you put the actual biochemical temperature effect. I am also confused why the ratio is always in the same direction. Why would forestation in Russia have a negative biochemical temperature impact?

Response: Thanks for your comment. We have rewritten this section. Please refer to the response above for the revision.

l. 245 – what about soil?

reforestation could potentially decrease soil carbon stocks (see, e.g., <https://doi.org/10.1046/j.1354-1013.2002.00486.x>)

In general, it is crucial that studies do not neglect soil, they often store more carbon than the biomass. I know this is not the main point of the paper, but I find it worrying that our community is writing paper after paper talking about changes in biomass and soil carbon is not mentioned even once. Please add some sort of discussion somewhere regarding soil carbon. Since you use the method of Windisch2019, why not include their soil aspect as well?

Response: Thanks for raising this important issue regarding soil carbon. We admit that soil carbon is a crucial part when evaluating the biochemical effect of land cover changes. We did not include soil carbon like Windisch et al., because their analysis of relative SOC variation is based on the type conversion from forest to non-forest type (e.g. conversion of forest to cropland decreases SOC by $26.6\% \pm 28.7\%$) (Sanderman et al., 2017), rather than the fractional tree cover changes. Since satellite observations do not provide direct estimations of soil carbon, we cannot obtain the soil carbon sensitivity to tree cover using the “space-for-time” method. Moreover, a recent study suggests that under the global tree restoration scenario, biomass change, rather than soil carbon change, is the main contributor to carbon sequestration (Veldman et al., 2019), despite the large amount of carbon stored in global soil.

Reference:

Sanderman, J., Hengl, T., Fiske, G.J., 2017. Soil carbon debt of 12,000 years of human land use. *Proc. Natl. Acad. Sci. U. S. A.* 114, 9575 – 9580. <https://doi.org/10.1073/pnas.1706103114>

Veldman, J.W., Aleman, J.C., Alvarado, S.T., Anderson, T.M., Archibald, S., Bond, W.J., Boutton, T.W., Buchmann, N., Buisson, E., Canadell, J.G., Dechoum, M. de S., Diaz-Toribio, M.H., Durigan, G., Ewel, J.J., Fernandes, G.W., Fidelis, A., Fleischman, F., Good, S.P., Griffith, D.M., Hermann, J.-M., Hoffmann, W.A., Le Stradic, S., Lehmann, C.E.R., Mahy, G., Nerlekar, A.N., Nippert, J.B., Noss, R.F., Osborne, C.P., Overbeck, G.E., Parr, C.L., Pausas, J.G., Pennington, R.T., Perring, M.P., Putz, F.E., Ratnam, J., Sankaran, M., Schmidt, I.B., Schmitt, C.B., Silveira, F.A.O., Staver, A.C., Stevens, N., Still, C.J., Strömberg, C.A.E., Temperton, V.M., Varner, J.M., Zaloumis, N.P., 2019. Comment on “The global tree restoration potential.”

Following your suggestion, we have added descriptions about soil carbon in the method section **(Line 542)**:

Notably, $\delta CO_2 e^{bchem}$ provides a simple estimation of the ideal carbon stock in biomass under current climate and disturbance regimes for further comparison with the biophysical effect. The period for restored forests to reach such carbon potential, and the role of changing climate and soil carbon flux in this process are neglected.

We also added discussions about the impact of tree restoration on soil carbon **(Line 323)**:

In addition, potential variations in soil carbon are ignored in the biochemical part. In contrast to increases in biomass, tree restoration could have positive and negative effects on soil carbon, depending on the climate background and the ecosystem type^{58–60}. At the global scale, forestation can increase soil organic carbon, but the value is highly uncertain⁶¹. Neglecting the potential change in soil carbon may lead to a slight underestimation of the biochemical effect⁶². The evaluated change in biomass should be the main contributor to carbon sequestration⁶³.

Reference:

58. Deng, L., Zhu, G. yu, Tang, Z. sheng & Shangguan, Z. ping. Global patterns of the effects of land-use changes on soil carbon stocks. *Glob. Ecol. Conserv.* 5, 127 – 138 (2016).
59. Sanderman, J., Hengl, T. & Fiske, G. J. Soil carbon debt of 12,000 years of human land use. *Proc. Natl. Acad. Sci. U. S. A.* 114, 9575 – 9580 (2017).
60. Guo, L. B. & Gifford, R. M. Soil carbon stocks and land use change: A meta analysis. *Glob. Chang. Biol.* 8, 345 – 360 (2002).
61. Mo, L. et al. Integrated global assessment of the natural forest carbon potential. *Nature* 624, 92 – 101 (2023).
62. Li, Y. et al. Prioritizing Forestation in China Through Incorporating Biogeochemical and Local Biogeophysical Effects. *Earth’ s Futur.* 12, 1 – 18 (2024).
63. Veldman, J. W. et al. Comment on “The global tree restoration potential” . *Science (80-.).* 366, 1 – 5 (2019).

1. 248 – the -0.11K are now for air temperature?

Response: Thanks for your comment. We have rewritten this section. Please refer to the response above for the updated content.

1. 251 – what do you mean with override? I think you mean a different word here. “exceed”, maybe?

Response: Thanks for your comment. We have rewritten this section. Please refer to the response above for the updated content.

Discussion

1. 1, 281 – should prob cite Winckler2019 here because the non-local effects can

be even larger than local effects

Response: Thank you for your professional suggestion. We have added a sentence about the non-local effect and cited the reference **(Line 309)**:

The non-local effect of forestation can even exceed the local effects in model simulations¹⁵.

Reference:

15. Winckler, J., Lejeune, Q., Reick, C. H. & Pongratz, J. Nonlocal Effects Dominate the Global Mean Surface Temperature Response to the Biogeophysical Effects of Deforestation. *Geophys. Res. Lett.* **46**, 745–755 (2019).

295 – good that you provide this explanation.

Response: Thanks for the positive comment.

1. 302 – but they are not that substantial anymore when considering T_a instead of T_s right? Generally, I find the wording a bit too positive. Your paper shows that the climate impact of tree planting is lower when considering the air temperature instead of the surface temperature.

Response: Thank you for the valuable suggestion. We have updated our results of comparison between T_s-based and T_a-based biophysical effects (please refer to the response above). We discussed the overestimation of biophysical climate effects based on T_s **(Line 299)**:

Through the comparison of biophysical and biochemical effects, we find that using T_s as the indicator may overestimate the role of biophysical processes in the overall climate effect of forestation. The evaluation based on the more relevant T_a can present better policy guidance for prioritizing the location of forestation.

We also discussed the application for different temperature metrics **(Line 329)**:

In the context of global climate change, T_s and T_a show comparable variation patterns and trend values³⁷. However, when assessing the temperature effects of afforestation or deforestation, the T_s-based values can be about five times higher than the T_a-based values. This significant difference in magnitude highlights that attention should be given to the evaluated temperature metrics and the application scenarios when interpreting the biophysical effects of land cover changes. For instance, T_s (i.e. canopy temperature) could be the more appropriate metric when considering the effects of biophysical processes on ecosystem metabolism of photosynthesis, respiration, and transpiration^{38,64}. Meanwhile, the more relevant T_a should be used in analyses related to regional climate adaptation of tree restoration. We highlight that this issue should be considered in future RS-based studies focusing on the thermal buffering effects of forests.

Reference:

37. Wang, Y. R., Hesse, D. O., Samset, B. H. & Stordal, F. Evaluating global and regional land warming trends in the past decades with both MODIS and ERA5-Land land surface temperature data. *Remote Sens. Environ.* **280**, 113181 (2022).
38. Mildrexler, D. J., Zhao, M. & Running, S. W. A global comparison between station

- air temperatures and MODIS land surface temperatures reveals the cooling role of forests. J. Geophys. Res. Biogeosciences 116, 1–15 (2011).*
64. Guo, Z. et al. Does plant ecosystem thermoregulation occur? An extratropical assessment at different spatial and temporal scales. *New Phytol.* (2022) doi:10.1111/nph.18632.

The discussion is missing the important point that forestation and forestry in general have numerous aspects outside of climate change mitigation that should not be forgotten over their climate impact because this can lead to bad strategies. The Bastin paper that you also base your research on was in part wrongly interpreted by the media, public, and politicians. It was understood in a way that we can simply plant our way out of the crisis. I therefore find it very important to briefly address this.

Some examples that come to mind, no need to add them all, but just to point you to some aspects:

- in many places, forests are not the ecosystem that “should be there” (see important work by Bond: <https://10.1016/j.tree.2019.08.003> , <https://10.1126/science.aad5132>)
- forests offer many ecosystem services, not only climate change mitigation
- tree planting can have negative ecological and economical impacts (e.g., <https://doi.org/10.1126/science.abd3064> and <https://10.1126/science.aac9892>)

Response: Thanks for the professional and helpful comments. We have added discussions about the widespread impacts of forestation (**Line 360**):

Moreover, tree restoration can have numerous ecological, hydrological and economic impacts besides the assessed temperature effects. Restoration in inappropriate geolocations (e.g. tropical savannas) can have counterproductive consequences such as ecosystem degradation, biodiversity loss and water availability reduction⁶⁸⁻⁷². These impacts should be also taken into account in the development of comprehensive forest-related strategies to avoid the misconception that “restoring trees is the panacea for the current crisis”.

References:

68. Bond, W. J., Stevens, N., Midgley, G. F. & Lehmann, C. E. R. The Trouble with Trees: Afforestation Plans for Africa. *Trends Ecol. Evol.* 34, 963–965 (2019).
69. Bond, W. J. Ancient grasslands at risk. *Science* (80-.). 351, 120–122 (2016).
70. Gómez-González, S., Ochoa-Hueso, R. & Pausas, J. G. Afforestation falls short as a biodiversity strategy. *Science* (80-.). 368, 1439–1439 (2020).
71. Parr, C. L., te Beest, M. & Stevens, N. Conflation of reforestation with restoration is widespread. *Science* (80-.). 383, 698–701 (2024).
72. Selva, N., Chylarecki, P., Jonsson, B.-G. & Ibisch, P. L. Misguided forest action in EU Biodiversity Strategy. *Science* (80-.). 368, 1438–1439 (2020).

Methods

The types of trees matter a lot, how is that accounted for? For instance, in temperate forests, if you reforest with broad-leaves, this will be much different than reforesting with needleleaves, especially regarding surface roughness in winter. How is that accounted for?

Response: Thanks for this professional comment. Since the temperature sensitivity is estimated by the local regression between tree cover and temperature observations, our results reflect the potential consequence of tree restoration with the native tree species. We have added descriptions in the revised manuscript (**Line 435**):

Notably, all the sensitivity results should be interpreted as the temperature consequences of restoration with native forest type, as the gridded tree cover data of existing species are used as inputs to the spatial regression model.

1. 321 – soil is 250m how was this aggregated to the 1km?

Response: Thanks for the comment. This paragraph, describing how to estimate tree cover potential (ΔTC) using soil and climate data, is removed, since the tree cover potential related contents are no longer presented in the revised manuscript according to the comment of reviewer #3.

1. 326 – Fig S8 – why is there only one such figure as there are two TC₂₀₂₀ datasets? Also, which of the two is shown?

Response: Thanks for raising this concern. In the original manuscript, we use tree cover data of 2020 to calculate tree restoration potential (ΔTC) under the current stage. According to the comment of reviewer #3, the analysis based on ΔTC is removed, and only tree cover data of 2010 are used for estimating the sensitivities (**Supplementary Fig. 12**).

Fig. S12. Global maps of fractional tree cover from (a) GLOBMAP and (b) Global

Forest Change (GFC) datasets.

1. 339 – mention that screen height is 1.25m to 2m above ground, I had to look this up.

Response: Thank you for the helpful suggestion. We have declared the physical meaning of assessed T_a here (**Line 397**):

The other temperature metric used for assessment is T_a , the air temperature at 2 m above the interface layer between the land components and the atmosphere. Specifically, for forested areas, the reference plane is the canopy, whereas for openlands, the reference plane is approximately the ground.

1. 342 – can you provide a figure of the modeled vs observed T_a

Response: Thanks for the suggestion. We have added the comparison between the satellite monthly mean T_a and FLUXNET monthly mean T_a measurements (**Line 406**):

Satellite monthly mean T_a and FLUXNET monthly mean T_a show good agreement. Validation results for forest and non-forest sites show comparable accuracy (Supplementary Fig. 13).

Fig. S13. Validation of satellite monthly mean air temperature (T_a) of 2010. (a) Scatter plot between satellite monthly T_a and FLUXNET monthly T_a measurements in forest sites. (b) Similar to (a), but for non-forest sites. Abbreviation: RMSE, root mean squared error.

1. 345 – mean = average of min and max? this seems quite odd to me. Especially with strong differences in daylength vs nightlength this does not seem like a valid way to get the mean temperature.

Response: Thanks for your comment. We compared the FLUXNET observations of (a) the mean of maximum and minimum T_a and (b) the daily mean T_a (Fig. R1). The comparison is performed at the monthly scale. The results show good overall agreement in either forest or non-forest sites (RMSE < 0.4 K). In addition, for both cold months (with longer night length) and hot months (with longer day length), the scatter plots are also close to the 1:1 line, suggesting the method is robust to different day/night length conditions.

Fig. R1. Comparison between max and min air temperature (Ta) and daily mean Ta using FLUXNET observations for (a) forest sites and (b) non-forest sites at the monthly scale.

1. 353 - is it really moving window? Or is the globe just gridded into 0.25° pixels? If it is moving window, what is the step size?

Response: Thanks for pointing this out. The regression is conducted for each 0.25° grid. We have removed the sentence about the moving window to avoid ambiguity.

1. 355 what is the impact? how many pixels are excluded? If it's a lot then this has an important impact on the importance of the results.

Response: Thank you for raising this concern. First, we set the threshold of tree cover $> 10\%$ for selecting the effective pixel for regression. This is because the temperature variations in low tree cover pixels may be induced by the rest non-forest land cover components. Using these pixels for regression may reduce the reliability of estimated temperature sensitivity. Second, for the pixels with tree cover $> 10\%$, we exclude part of them with water cover $> 1\%$ and elevation difference > 100 m. Here, we show the ratio of the excluded forest pixel number (due to water cover or elevation difference) to the total forest pixel number within each 0.25° grid (Fig. R2). Results show that the excluded pixel numbers are small across the globe ($< 5\%$ for most grids). Thus, the screening process may have limited impacts on our sensitivity results.

Fig. R2. Fraction of excluded pixels due to water cover or elevation differences, defined as the number of abandoned forest pixels (tree cover $> 10\%$) divided by the number of all forest pixels within each 0.25° grid.

1. 357 I think there is an "and" missing

Response: Thanks for your careful reading. We have added the word "and" (**Line 420**):

The water coverage and elevation data are from Joint Research Center Global Surface Water Mapping Layers v1.4, and GMTED2010 datasets, respectively.

1. 361: for each 0.25 pixel, right?

Response: Thanks for pointing this out. We added the clarification about the scale (**Line 424**).

After the screening process, δT_s^{bph} and δT_a^{bph} can be estimated using a linear regression model between tree cover and corresponding temperature values for each 0.25° grid:

Formulas 1+2: Why not just write the formula of the linear model instead of the solution of the regression?

Response: Thanks for the helpful suggestion. We have revised the formulas for clarity (**Line 425**):

$$T_s = \delta T_s^{bph} \times TC_{2010} + b_s \quad (1)$$

$$T_a = \delta T_a^{bph} \times TC_{2010} + b_a \quad (2)$$

where, b_s and b_a are the regression intercepts.

Formulas 1+2: provide a plot for a pixel in the supplements, to understand the relationship

Response: Thank you for the valuable comment. We have added the schematic figure as you suggested (**Supplementary Fig. 1**).

Fig. S1. Schematic representation of the methodology for estimating land surface temperature (T_s) or air temperature (T_a) sensitivity to tree cover change (δT_s^{bph} and δT_a^{bph}). (a–f) The example grid (59.75°–60°N, 63.5°–63.75°E) with positive δT_s^{bph} and δT_a^{bph} in January 2010. (a) True color image of the 0.25° grid. (b) GLOBMAP 2010 tree cover map. (c) Mean T_s of January 2010. (d) Mean T_a of January 2010. (e) Linear regression for estimating positive δT_s^{bph} of the grid. (f) Linear regression for estimating positive δT_a^{bph} of the grid. (g–l) Similar to (a–f), but for the other example grid (44.75°–45°N, 38°–38.25°E) with negative sensitivities in July 2010.

l. 364 – what robustness? you mean that the formula is solvable, i.e., the matrix is invertible?

Response: Thanks for the comment. The small sample sizes may result in: (a) models that are sensitive to noise of the input data, and (b) estimated parameters that are unstable or have large uncertainties. We have revised the description for clarity (**Line 427**):

To ensure the reliability of the results,

l. 383 – “of” missing

Response: Thanks for the careful reading. We have added the missing “of” (**Line 450**):

The RS-based local temperature sensitivity can be validated through the differences of measurements between spatially adjacent paired forest and non-forest sites.

1. 390 – BEST is at 1 degree resolution. So it's not only land cover type impact but also elevation impact. How is that accounted for?

Response: Thanks for the constructive comment. We have made the additional elevation correction for the gridded temperature using (a) the difference between grid mean and corresponding site elevations, and (b) the lapse rate of the grid. The lapse rate is estimated by the local regression between the gridded temperatures and elevations within 5×5 window. Due to modifications made to the methodology, all the related results are also updated, and the systematic biases in site results are eliminated. We also revised descriptions for the elevation correction (**Line 458**):

Here, we first make corrections to the gridded temperatures using the lapse rates to compensate for the elevation difference between the site and the corresponding grid. The lapse rate for the target grid is estimated by the regression slope of the gridded temperatures and elevations within the 5×5 window. Then, by deducting the corrected gridded temperature data, the in-situ measurements can effectively represent the relative temperature effects attributable to different land cover types.

1. 446 – why to 2m above the canopy? Isn't the goal to estimate 2m above ground? A figure would generally be good to visualize the difference in the temperatures...

Response: Thanks for the comment. The target variable is the 2 m air temperature above the land surface. We have added a schematic figure about the temperature metrics (**Supplementary Fig. 2**). Please refer to the above responses for the definition and the importance of the evaluated temperature.

Fig. S2. Schematic representation of evaluated land surface temperature (T_s) and near surface air temperature (T_a).

1. 471 – I am unsure how Fig S6 relates to that. Instead, can you provide a figure comparing the non-gap-filled and the gap-filled maps?

Response: Thanks for the comment. According to the comment of reviewer #3, the analyses based on gap-filled sensitivity and the related figures are removed.

1. 474 – why is this aggregation needed?

Response: Thanks for the comment. This sub-section about sensitivity gap-filling is removed.

1. 480 – how did you account for spatial auto-correlation? Training and testing data need to be independent, see <https://www.nature.com/articles/s41467-022-29838-9>

Response: Thanks for the comment. This sub-section about sensitivity gap-filling is removed.

1. 504 – this is missing soil carbon

Response: Thanks for raising this important issue. We have clarified in the methodology that soil carbon is ignored and discussed the possible consequences in the discussion section. Please refer to the response above for the modifications.

Code Availability

Why is the code available only on request? This should be made public, we need to make results reproducible.

Response: Thanks for raising this important issue. We have provided a Zenodo link to download the code (Line 561):

Code availability

The Python codes used to generate all the results are available at <https://zenodo.org/uploads/14215398>

Reviewer #2 (Remarks on code availability):

The code is hard to review. There are no comments or explanations.

Response: Thanks for the careful reading. We have added a description file and comments to the code.

What I found strange was, that the code to reproduce Figure 1 only reads in images that contain the temperature-forestcover relation and plots them. But there is no code that created these images. I assume these were generated in GEE, but this code should also be made available.

Response: Thanks for the comment. We have uploaded the code file for the sensitivity estimation.

Reviewer #3 (Remarks to the Author):

This paper combines satellite data, and data from meteorological station and flux tower networks, to explore the extent to which reforestation confers a local air cooling benefit across the globe. We have long known that reforestation leads to local surface cooling, at least in the tropic and temperate zones. So this adaptive benefit of reforestation is not exactly “overlooked” as the title of the paper suggests. However, because air temperature can not be directly measured from space, studies on the topic have tended to focus on patterns in surface temperature only, despite the fact that air temperature is arguably the more relevant target for adaptation. The authors acknowledge that some work has already attempted to address this knowledge gap, but I think that what the authors present in this paper is still a novel contribution. They show that forests likely lead to local air cooling, but the magnitude of the temperature change is less than the change in surface temperature. This is consistent with the other work done on the topic, including the nice study by Mildrexler et al. 2011 (which the authors don't cite but probably should, <https://doi.org/10.1029/2010JG001486>).

I did, however, have some significant comments and concerns about the framing and the methods.

Response: We appreciate the positive comments by the reviewer. We have carefully considered the critical comments, please see our detailed point-by-point responses below. In particular, we have updated the results and clarified the new contribution of this paper. Accordingly, we have revised the title to “Observed different impacts of potential tree restoration on local surface and air temperature”. We also cited the reference as you suggested (Line 66):

Despite the strong correlation between T_s and T_a , the T_s effect of forest change may significantly differ from the T_a effect³⁸.

38. Mildrexler, D. J., Zhao, M. & Running, S. W. A global comparison between station air temperatures and MODIS land surface temperatures reveals the cooling role of forests. *J. Geophys. Res. Biogeosciences* **116**, 1–15 (2011).

[1] First, and perhaps most importantly, the air (or surface cooling) benefits of reforestation should not be referred to as “climate mitigation.” Climate mitigation refers to efforts to reduce emissions and increase sinks for greenhouse gases. The direct modification of local temperature by forest restoration does neither of these. It can slow the pace of climate change in certain places by increasing sensible and latent heat fluxes. However, these mechanisms may simply represent the redistribution of energy within the climate system, and have the potential to enhance the pace of warming elsewhere. For example, energy used to evaporate water in a forested landscape is re-released when that water vapor condenses downwind. Likewise, greater sensible heat flux may increase lower temperatures near the surface but increase temperatures higher in the boundary layer. Thus, the direct local cooling benefits of reforestation

should be described as “adaptation,” not “mitigation,” and discussion of the benefits should acknowledge the possibility for warming elsewhere.

Response: Thanks for this critical comment. We have checked through our manuscript and replaced “mitigation” with “adaption” when discussing the local biophysical effect on climate. We also highlighted that our results should be related to adaption rather than mitigation in the discussion section (**Line 355**):

From the perspective of the whole climate system, the non-radiative effects represent the redistribution of energy within the climate system and may lead to warming in downwind regions or the higher boundary layer⁶⁷. Thus, our results concerning the biophysical effects should be treated as the reference for local climate adaptation rather than global climate mitigation.

67. Barnes, M. L. et al. A Century of Reforestation Reduced Anthropogenic Warming in the Eastern United States. *Earth's Futur.* 12, (2024).

[2] For this reason, it is also inappropriate to directly compare the change in LOCAL surface or air temperature associated with reforestation to the GLOBAL change in air temperature driven by the capacity of forests to enhance the global land carbon sink (specifically, the delta_TC metric which the authors present in Figure 4). They are not the same thing. Moreover, because the methods the authors used to calculate delta_TC do not appear to have been published elsewhere and are not clearly validated, I suggest the authors remove this part of the analysis from the manuscript.

Response: Thanks for this critical comment. As you suggested, we have removed the analyses based on ΔTC . We have rewritten this section by comparing the concept of sensitivity (**Line 245**). The revision makes our results clearer that only local effects are evaluated and the non-local or global effects are not included. Similar to the previous study (Windisch et al., 2021), the current comparisons are based on the equivalent CO₂ uptake, rather than the temperature effect (K). The new results allow us to investigate the relative importance of the biophysical effect and reveal the overestimation of the biophysical climate effect using T_s (rather than the more relevant T_a) as the metric.

References:

Windisch, M. G., Davin, E. L. & Seneviratne, S. I. Prioritizing forestation based on biogeochemical and local biogeophysical impacts. *Nat. Clim. Chang.* 11, 867 – 871 (2021).

Comparison of biophysical with biochemical effects based on two temperature metrics

Most assessments of the climate benefits related to forestation have concentrated on the carbon sequestration (i.e., biochemical effects)^{52,53}. Here, the biomass carbon stock sensitivity to tree cover is estimated via space-for-time analogy and converted to CO₂ absorption equivalents ($\delta CO_2 e^{bchem}$) to represent the biochemical effect. We also convert the biophysical T_s and T_a sensitivities to the metric of equivalent CO₂ uptake ($\delta CO_2 e^{bph,T_s}$ and $\delta CO_2 e^{bph,T_a}$, Supplementary Fig. 11). These allow the

comparison of the local biophysical and biochemical climate effects and evaluation of the relative importance of the former.

The spatial map shows that δCO_2e^{bchem} in tropical rainforests can exceed 600 t/ha, which is much greater than temperate and boreal forests (Fig. 4a). This suggests that restoring damaged forests in tropical regions has the most carbon benefit. Latitudinally, δCO_2e^{bchem} at low-latitudes is higher than that at mid- or high-latitudes, with global mean of 268.2 ± 37.8 t/ha (mean \pm uncertainty) (Fig. 4b). In terms of the biophysical effect, $\delta CO_2e^{bph,Ts}$ (41.7 ± 9.3 t/ha) provides a global average of 15.7% additional benefits to δCO_2e^{bchem} (Fig. 4b). However, if the more relevant biophysical Ta effect is considered, the ratio of $\delta CO_2e^{bph,Ta}$ (9.3 ± 2.9 t/ha) to δCO_2e^{bchem} is only 3.5%.

We then focus on northern high-latitudes, where tree restoration shows a biophysical warming effect. The resultant negative $\delta CO_2e^{bph,Ts}$ could offset 9.5% of the δCO_2e^{bchem} annually (Fig. 4b). The high-latitude biophysical warming is more pronounced in the cold season and can reduce the biochemical climate effect by 42.4% in March (Fig. 4c). However, when $\delta CO_2e^{bph,Ta}$ is used as the indicator, the offset of biophysical to biochemical effects is only 3.3% at the annual scale, with the maximum monthly value of 10.6% (February) (Fig. 4b and 4c). In mid-latitudes, the seasonal $\delta CO_2e^{bph,Ts}$ can enhance δCO_2e^{bchem} by up to 33.7% (northern hemisphere) and 40.5% (southern hemisphere) during summer. However, these seasonal ratios are only about 10% considering $\delta CO_2e^{bph,Ta}$ (Fig. 4d and 4f). In low-latitudes, annual positive $\delta CO_2e^{bph,Ts}$ is equivalent to 25.5% of δCO_2e^{bchem} , while the ratio for $\delta CO_2e^{bph,Ta}$ is only 6.2%, with insignificant seasonal variations (Fig. 4b and 4e). These results suggest that the relative importance of biophysical effects largely depends on the evaluated temperature metric, and the role of biophysical effects in the overall climate effect (usually measured by Ta) may not be as important as estimated in previous Ts-based studies^{22,23}.

Figure 4. Comparison of the biophysical (bph) and biochemical (bchem) effects of potential tree cover gain. (a) Global pattern of the biochemical effect of potential tree cover gain (δCO_2e^{bchem}). (b) Global and latitudinal means of biochemical and biophysical effects of potential tree cover gain. The Ts-based and Ta-based biophysical effects are

shown as the equivalent CO₂ uptake ($\delta CO_2 e^{bph,Ts}$ and $\delta CO_2 e^{bph,Ta}$). The error bars indicate the uncertainty of the mean. (c–f) Monthly ratios of Ta-based and Ts-based biophysical effects to equivalent biochemical effects across northern high-latitudes (>50°N), northern mid-latitudes (20°–50°N), tropics (20°S–20°N) and southern mid-latitudes (> 20°S). The shaded area indicates the uncertainty of the ratios.

We also rewrite the corresponding method section (**Line 535**):

Comparison of biophysical and biochemical effects

In addition to regulating the energy balance process, forestation can enhance the land carbon sink through vegetation photosynthesis, thereby generating negative biochemical feedback on the climate system^{92,93}. To quantify this biochemical impact, we first estimate the biomass carbon density sensitivity to ideal restoration, using Global Aboveground and Belowground Biomass Carbon Density Maps of 2010 (in t/ha)⁹³, along with TC_{2010} and the “space-for-time” strategy. We convert the biomass carbon stock sensitivity to CO₂ absorption equivalents (i.e. $\delta CO_2 e^{bchem}$) based on the molar mass ratio. Notably, $\delta CO_2 e^{bchem}$ provides a simple estimation of the ideal carbon stock in biomass under current climate and disturbance regimes for further comparison with the biophysical effect. The period for restored forests to reach such carbon potential, and the role of changing climate and soil carbon flux in this process are neglected.

The biophysical Ts and Ta sensitivities are also unified to the metric of CO₂ equivalents ($\delta CO_2 e^{bph,Ts}$ and $\delta CO_2 e^{bph,Ta}$) based on the transient climate response to cumulative emissions for both Ts ($TCRE^{Ts}$) and Ta ($TCRE^{Ta}$) derived from Coupled Model Intercomparison Project Phase 6 (CMIP6) simulations (Supplementary Fig. 11):

$$\delta CO_2 e^{bph,Ts} = \frac{\delta T_s^{bph}}{TCRE^{Ts}} \times \frac{1}{A_E} \quad (11)$$

$$\delta CO_2 e^{bph,Ta} = \frac{\delta T_s^{bph}}{TCRE^{Ta}} \times \frac{1}{A_E} \quad (12)$$

where, A_E indicates the earth surface area ($5.1 \times 10^8 \text{ km}^2$). The gridded $TCRE^{Ts}$ and $TCRE^{Ta}$ are estimated following the previous study²², using 12 model simulations (ACCESS_ESM1-5, CanESM5-1, CMCC-ESM2, CNRM-ESM2-1, FIO-ESM-2-0, GISS-E2-1-H, INM-CM5-0, IPSL-CM6A-LR, MIROC6, MPI-ESM1-2-LR, MRI-ESM2-0 and NESM3) of the “1 percent per year increase in carbon dioxide” experiment (1pctCO2). Notably, $\delta CO_2 e^{bph,Ts}$ and $\delta CO_2 e^{bph,Ta}$ calculated by equation (11) and (12) represent the CO₂ emission equivalents. We further covert their signs to align with the $\delta CO_2 e^{bchem}$, which represent the CO₂ absorption equivalents. We compare the biophysical and biochemical effects based on the above metrics at both annual and monthly scales.

References:

22. Windisch, M. G., Davin, E. L. & Seneviratne, S. I. Prioritizing forestation based

- on biogeochemical and local biogeophysical impacts. Nat. Clim. Chang.* **11**, 867–871 (2021).
23. *Zhu, L. et al. Comparable biophysical and biogeochemical feedbacks on warming from tropical moist forest degradation. Nat. Geosci.* **16**, 244–249 (2023).
52. *Walker, W. S. et al. The global potential for increased storage of carbon on land. Proc. Natl. Acad. Sci. U. S. A.* **119**, 1–12 (2022).
53. *Lewis, S. L., Wheeler, C. E., Mitchard, E. T. A. & Koch, A. Restoring natural forests is the best way to remove atmospheric carbon. Nature* **568**, 25–28 (2019).
92. *Xu, H., Yue, C., Zhang, Y., Liu, D. & Piao, S. Forestation at the right time with the right species can generate persistent carbon benefits in China. Proc. Natl. Acad. Sci.* **120**, 2017 (2023).
93. *Spawn, S. A., Sullivan, C. C., Lark, T. J. & Gibbs, H. K. Harmonized global maps of above and belowground biomass carbon density in the year 2010. Sci. Data* **7**, 1 – 22 (2020).

[3] Relatedly, the darkening of the land surface associated with the tendency for forests to lower albedo is a process that actually contributes to additional warming at regional and global scales, even if modifications to latent and sensible heat exchange tend to suppress temperature locally. This point, which is discussed in Barnes et al. (2024, <https://doi.org/10.1029/2023EF003663>) is important to make, because otherwise it is very hard to reconcile results from this paper with those that demonstrate that the climate mitigation potential of reforestation is reduced by the albedo impacts (including the paper by Hasler et al. published this year in Nature Communications, <https://doi.org/10.1038/s41467-024-46577-1>).

Response: Thanks for the constructive comment. We have added discussions about the albedo effect of forestation to reconcile with previous studies (**Line 350**):

The biophysical warming effects of boreal forests should be given specific attention in related mitigation policies, although our results of negative biophysical climate effects at high-latitudes may not be as strong as previous findings^{65,66}. This is because those studies focus on the additional radiative forcing induced by the darker forest canopy but ignore the impact of turbulent fluxes. The overlooked non-radiative effects could partially offset the albedo effects, leading to the observed net warming in our results. From the perspective of the whole climate system, the non-radiative effects represent the redistribution of energy within the climate system and may lead to warming in downwind regions or the higher boundary layer⁶⁷. Thus, our results concerning the biophysical effects should be treated as the reference for local climate adaptation rather than global climate mitigation. The fact that the mitigation potential of high-latitudes forestation could be reduced or even offset by the albedo impacts should be considered by forest-related global mitigation policies.

References:

65. Hasler, N. et al. Accounting for albedo change to identify climate-positive tree cover restoration. *Nat. Commun.* **15**, 2275 (2024).
66. Weber, J. et al. Chemistry-albedo feedbacks offset up to a third of forestation's CO₂ removal benefits. *Science* (80-.). **383**, 860–864 (2024).
67. Barnes, M. L. et al. A Century of Reforestation Reduced Anthropogenic Warming in the Eastern United States. *Earth's Futur.* **12**, (2024).

A few other comments:

[4] The authors focus very strongly on the impacts of reforestation on roughness and sensible heat flux, but seem to ignore the impacts on latent heat flux (e.g. evapotranspiration). Other work on the topic (including papers cited by the authors already) make it clear that both processes are important for governing the local temperature impacts of reforestation.

Response: Thanks for the professional comment. The enhancement in latent heat flux can strongly affect the local Ts effect of forestation at low-latitudes. We have added related discussions in the revised manuscript (**Line 287**):

Previous studies have demonstrated that in boreal regions, forests can warm local Ts because the tree canopy is darker than the snow background and absorbs more solar radiation; in tropical regions, forests show strong local Ts cooling, mainly due to the higher evapotranspiration rates than other vegetation or bare land; in temperate regions, the net Ts effect depends on the relative magnitude of these two processes^{10,12,29,31}.

References:

10. Lawrence, D., Coe, M., Walker, W., Verchot, L. & Vandecar, K. The Unseen Effects of Deforestation: Biophysical Effects on Climate. *Front. For. Glob. Chang.* **5**, 1–13 (2022).
12. Li, Y. et al. Local cooling and warming effects of forests based on satellite observations. *Nat. Commun.* **6**, 1–10 (2015).
29. Duveiller, G. et al. Biophysics and vegetation cover change: A process-based evaluation framework for confronting land surface models with satellite observations. *Earth Syst. Sci. Data* **10**, 1265–1279 (2018).
31. Bright, R. M. et al. Local temperature response to land cover and management change driven by non-radiative processes. *Nat. Clim. Chang.* **7**, 296–302 (2017).

[5] I appreciate that the authors engaged in a robust effort to use data from eddy covariance flux towers to validate their approach of comparing land surface temperature measurements from satellites with gridded information on air temperature from meteorological station networks. However, this effort seems to compare the flux tower-derived information to a gridded air temperature product derived from CRU and BEST data. Why didn't the authors compare flux tower data to the air temperature product they actually use for the global mapping in the study (e.g. the dataset produced by Zhang et al 2022)?

Response: Thanks for the professional comment. The CRU or BEST data used here are interpolated

temperature data, which represent the influence of the macroclimate. Therefore, their differences with site measurements reflect the relative influence of land cover on local temperature. Zhang et al's Ta data, on the other hand, are produced by a statistical method using satellite land surface temperature and thus contain both signals from macroclimate and land cover. The difference between Zhang et al. Ta data and site measurements may reflect systematic or random errors of the product rather than the information we are interested in.

[6] The presentation of equations [1] and [2] is confusing. Not all of the terms in these equations are defined, and the math seems to simply considerably from what is shown (for example, as written, it seems the TC_{2010}^T terms simply cancel out of the equations?)

Response: Thanks for raising this concern. The equations in the original manuscript are the solution of the linear model represented by the matrix operation form. The superscript “T” denotes the matrix transpose. We have revised the formulas for clarity (**Line 425**):

$$T_s = \delta T_s^{bph} \times TC_{2010} + b_s \quad (1)$$

$$T_s = \delta T_a^{bph} \times TC_{2010} + b_a \quad (2)$$

where, b_s and b_a are the regression intercepts.

[7] The relationship between surface temperature and outgoing long-wave radiation (equation 3) includes a correction for incident long-wave radiation. This is unusual...can the authors provide a reference or further justification for the approach?

Response: Thanks for the comment. Equation 3 ($T_s = \left[\frac{LW_u - (1-\varepsilon)LW_d}{\varepsilon\sigma} \right]^{\frac{1}{4}}$) has been widely used for land surface temperature retrieval from site longwave radiation measurements (Duan et al., 2019; Ma et al., 2021; Schultz et al., 2017). Here, the correction is made to exclude the reflected longwave radiation from the measured outgoing longwave radiation, and thereby obtain the actual value of emitted radiation.

References:

- Duan, S.B., Li, Z.L., Li, H., Göttsche, F.M., Wu, H., Zhao, W., Leng, P., Zhang, X., Coll, C., 2019. Validation of Collection 6 MODIS land surface temperature product using in situ measurements. *Remote Sens. Environ.* 225, 16 – 29. <https://doi.org/10.1016/j.rse.2019.02.020>
- Ma, J., Zhou, J., Liu, S., Göttsche, F.M., Zhang, X., Wang, S., Li, M., 2021. Continuous evaluation of the spatial representativeness of land surface temperature validation sites. *Remote Sens. Environ.* 265. <https://doi.org/10.1016/j.rse.2021.112669>
- Schultz, N.M., Lawrence, P.J., Lee, X., 2017. Global satellite data highlights the diurnal asymmetry of the surface temperature response to deforestation. *J. Geophys. Res. Biogeosciences* 122, 903 – 917. <https://doi.org/10.1002/2016JG003653>

We have added a reference for equation 3 (**Line 781**):

87. *Duan, S. B. et al. Validation of Collection 6 MODIS land surface temperature product using in situ measurements. Remote Sens. Environ. 225, 16 - 29 (2019).*

Response to Reviewers' Comments

We greatly appreciate the opportunity to revise our manuscript and thank all anonymous reviewers for their constructive comments. Below are the point-by-point responses to the comments, along with the revision of the manuscript (typed in Italics and Arial) and the location of the revision. The line numbers referred to are for the clean version of the revised manuscript.

Reviewer #1 (Remarks to the Author):

I thank the authors for the elaborate answers and thoughtful updates to the manuscript. I have no further comments.

Response: We appreciate the constructive and valuable comments by the reviewer during the review process.

Reviewer #1 (Remarks on code availability):

I could not access the code: "You do not have sufficient permissions to view this page."

Response: We apologize for the data accessibility issue. We have confirmed that the Zenodo project is publicly accessible (<https://zenodo.org/records/14633331>).

Reviewer #2 (Remarks to the Author):

The manuscript is now in much better shape, and I think it is a very interesting one and shows nice and relevant results. I especially like that the authors analyze the hypotheses both with remote sensing but also site information with FLUXNET data (they did so in the previous version already, but I still want to highlight this).

My concerns from the first round were thoroughly addressed, but a few issues remain that I believe require minor revisions from the authors.

Response: We thank the reviewer for the positive feedback. Based on the second-round comments, we mainly revised the methods section to make the manuscript clearer. Please see the point-by-point responses below.

Results

- I now fully understand what you mean by "100% tree cover gain" and Fig S1 makes it very clear, thank you. But I think the term is misleading/confusing. It should be something like "full tree cover" or "full restoration" or something like that. Because a 100% gain would mean, if I have 5 hectares of forest in some area, and then you plant 5ha more, that is a 100% gain in tree cover. But that does not mean that the area is fully restored aka has 100% tree cover. I think this needs to be termed differently.

Response: Thanks for pointing this out. We have revised "100% tree cover gain" to "the full tree cover restoration" to avoid ambiguity (**Line 76**).

We first estimated the local biophysical T_s and T_a sensitivity to the full tree cover restoration (denoted as δT_s^{bp} and δT_a^{bph}) at the 0.25° scale, based on the space-for-time analogy.

We have also revised the figure caption (**Line 124**).

Figure 1. Annual mean temperature sensitivity to the full tree cover restoration.

Methods

The procedure for the FLUXNET sites is now better explained, but still unclear.

Response: Thanks for raising this concern. We rearranged the method section and hope the revision can make the procedure clearer (**Line 464**):

The specific process of validation is as follows (Supplementary Fig. 8). In-situ data for T_a are measured above the vegetation canopy, whereas T_s is estimated using the longwave radiation measurements:

$$T_s = \left[\frac{LW_u - (1 - \varepsilon)LW_d}{\varepsilon\sigma} \right]^{\frac{1}{4}} \quad (3)$$

where, LW_u and LW_d represent upward and downward longwave radiation from the FLUXNET2015 dataset, respectively; σ denotes the Stephan-Boltzmann constant ($5.67 \times 10^{-8} \text{ W m}^{-2} \text{ K}^{-4}$), and ε is emissivity, estimated based on an empirical relationship with albedo⁸⁸. For the gridded data, we first make corrections using the

lapse rates to compensate for the elevation difference between the site and the corresponding grid. The lapse rate for the target grid is estimated by the regression slope of the gridded temperatures and elevations within the 5×5 window.

By deducting the corrected gridded temperature data, the in-situ measurements can effectively represent the land cover impacts on local T_s and T_a , assuming that macroclimate affects both temperature metrics similarly. Since the forest data cannot be directly matched with the openland data, we bin both forest and openland data points using the SW_d interval of 10 w/m^2 . For each SW_d bin, we calculate the difference between mean values of forest and openland data points to represent the temperature effect of forestation (i.e. δT_s^{bph} or δT_a^{bph*}) under the specific radiation background:*

$$\delta T_s^{bph*} = \left(\overline{T_s^{site} - T_f^{grid}} \right) - \left(\overline{T_s^{site} - T_o^{grid}} \right) \quad \text{if } SW_d \in (10k, 10k + 10) \quad (4)$$

$$\delta T_a^{bph*} = \left(\overline{T_a^{site} - T_f^{grid}} \right) - \left(\overline{T_a^{site} - T_o^{grid}} \right) \quad \text{if } SW_d \in (10k, 10k + 10) \quad (5)$$

Here, T_s^{site} and T_a^{site} refer to T_s and T_a measured at forest sites, respectively; T_s^{site} and T_a^{site} refer to T_s and T_a measured at openland sites; T_f^{grid} and T_o^{grid} refer to the corresponding gridded temperatures after the elevation correction; k indicates counting of the SW_d bin. According to the metadata of the FLUXNET2015 dataset, forest sites include the following four IGBP landcover types: evergreen needleleaf forests, evergreen broadleaf forests, deciduous broadleaf forests, and mixed forests; openland sites are categorized as other non-forest vegetation types.

Then, the relationships between two temperature sensitivities and SW_d are explored using the weighted least squares (WLS) regression model, in which the samples are δT_s^{bph} or δT_a^{bph*} of all SW_d bins and the sample weights are defined as the inverse of the standard error of δT_s^{bph*} or δT_a^{bph*} . The derived relationships are then compared with those from RS-based results for validation. Here, the monthly ERA5-Land shortwave radiation data are used to build the relationships with RS-based sensitivities. We also compare and validate the maximum temperature sensitivities.*

- why is the binning necessary for the fluxnet sites? I looked at the data in the supplements and it seems you could just provide a plot with all data points as well.

Response: Thanks for the valuable comment. First, we should note that the binning process is for all observations of forest (or openland) based on the shortwave radiation condition. In other words, the binning process is for observations rather than sites.

Since the forest and openland observations may be spatially distant or temporally asynchronous, they cannot be directly matched with each other (here the data indicate the corrected FLUXNET observations by subtracting gridded temperature). Thus, the binning process is necessary to provide a similar radiation background for comparing the “temperature effects” of forests and openlands to represent the consequence of land cover conversion. We have added a sentence to illustrate the reason for the binning process (**Line 476**):

Since the forest data cannot be directly matched with the openland data, we bin both forest and openland data points using the SW_d interval of 10 w/m^2 .

- In equation 4 and 5, what is the mean taken over? is it the average over all forest sites, and then the average over all openland sites? This is not written out in a mathematically clear way and so I don't understand it.

Response: Thanks for the valuable comment. The mean is taken over for all forest (or openland) observations within each shortwave radiation bin. We have revised the equations 4 and 5 for clarity **(Line 477)**:

For each SW_d bin, we calculate the difference between mean values of forest and openland data points to represent the temperature effect of forestation (i.e. δT_s^{bph} or δT_a^{bph*}) under the specific radiation background:*

$$\delta T_s^{bph*} = \left(\overline{T_s^{site} - T_f^{grid}} \right) - \left(\overline{T_s^{site} - T_o^{grid}} \right) \quad \text{if } SW_d \in (10k, 10k + 10) \quad (4)$$

$$\delta T_a^{bph*} = \left(\overline{T_a^{site} - T_f^{grid}} \right) - \left(\overline{T_a^{site} - T_o^{grid}} \right) \quad \text{if } SW_d \in (10k, 10k + 10) \quad (5)$$

- why is it called a relative effect (e.g. in figure S8). You're doing a simple subtraction, so that's an absolute effect, it's not put in relation to anything else.

Response: Thanks for the careful reading. We have revised the relative effect to “land cover impacts on Ta/Ts” for clarity **(Line 474)**:

By deducting the corrected gridded temperature data, the in-situ measurements can effectively represent the land cover impacts on local Ts and Ta

We also modified the wording in fig.S8 by using the “quantitative metric of forest (openland) impacts on local Ta or Ts”.

Supplementary Figure 8. Flow chart of the remote sensing (RS)-based local temperature effect validation using FLUXNET and gridded temperature data.

- It is unclear for what values the regression is done. For all of the datapoints? For the means of the bins?

Response: Thanks for the comment. The means of bins are used for regression. We have highlighted the regression data as you suggested (**Line 489**):

Then, the relationships between two temperature sensitivities and SW_d are explored using the weighted least squares (WLS) regression model, in which the samples are δT_s^{bph} or δT_a^{bp*} of all SW_d bins and the sample weights are defined as the inverse of the standard error of δT_s^{bp*} or δT_a^{bph*} .*

Fig S11 has inconsistent axes in a) and b), and a typo for b) in the caption (should be Ta)

Response: Thanks for the careful reading. We have redrawn the supplementary figure with the same color bar and revised the figure caption.

Supplementary Figure 11. Equivalent CO₂ uptake induced by biophysical Ts and Ta effects of potential tree restoration ($\delta\text{CO}_2e^{\text{bph},T_s}$ and $\delta\text{CO}_2e^{\text{bph},T_a}$). (a) Global map of $\delta\text{CO}_2e^{\text{bph},T_s}$. (b) Global map of $\delta\text{CO}_2e^{\text{bph},T_a}$. (c) Transient climate response to cumulative emissions (TCRE) of CO₂ for annual Ts. (d) Transient climate response to cumulative emissions (TCRE) of CO₂ for annual Ta.

Just a suggestion: Put the 600 tC/ha in context, e.g. with Pan2024, stating how dense those biomes are currently (around 240 tC/ha if I recall correctly)

Response: Thanks for the valuable comment. We should first note that the figure is δCO_2e , which indicates the equivalent CO₂ densities (t/ha), rather than C densities.

According to extended data table 2 of Pan et al., the mean biome-level C density of tropical intact forest is 293.1 tC/ha. Considering the ratio of living biomass to total C (living biomass, dead wood, litter and soil) derived from the table data, the estimated mean biomass C density is about 164.2 tC/ha. After converting the unit to CO₂, the number is about 602.2 tCO₂/ha, which is comparable to our result. Notably, we use the data of tropical intact forest rather than the tropical regrowth forest for comparison. This is because our density results are based on the full tree cover restoration, which is closer to the state of intact forest.

Following your suggestion, we have added the context of our results (**Line 253**):

The spatial map shows that $\delta\text{CO}_2e^{\text{bchem}}$ in tropical rainforest margins can exceed 600 t/ha (Fig. 4a), which is comparable to the previous estimation of tropical intact forests based on ecological research network observations⁵. This value is greater than $\delta\text{CO}_2e^{\text{bch}}$ in temperate and boreal forests, suggesting the highest carbon benefit of restoring damaged or degraded tropical forests.

5. Pan, Y. et al. The enduring world forest carbon sink. *Nature* 631, 563–569 (2024).

Reviewer #2 (Remarks on code availability):

The code requires a small effort. First of all, the packages and versions with which to run this are not included. So I had to manually install packages one by one which is annoying.

Response: Thanks for raising this concern. We have updated the Zenodo project (<https://zenodo.org/records/14633331>). The “environment.zip” file is uploaded, which contains the

“yml” file and the python wheels of GDAL for both Linux and Windows systems. We also updated the “readme.md” file, which provides simple instructions about the environment installation.

You should provide a conda environment.yml or something like that so one can just set up the environment and re-run the code. I managed to get the code running after I fixed some issues.

Response: Please refer to the response above. The “yml” file is uploaded.

BTW, the code does not directly run on Linux/Mac because of Windows-style file paths and sometimes wrong capitalization of filenames (that matters on Unix systems). Also, the names of the imports lib_image are wrong (should be function_image).

But I was able to reproduce the figures in the end.

Response: We apologize for the bugs in the code. We have checked through the code and debugged on both Windows and Linux.

Reviewer #3 (Remarks to the Author):

The authors have done a thorough job responding to comments raised during the first round of review. My particular concerns have been adequately addressed, and I do not have any others to raise at this point.

Response: We thank the reviewer for the constructive comments during the review process.